# Differentiated Aggregation to Improve Generalization in Federated Learning

**Peyman Gholami**                                                                        *pghola2@uic.edu*
*Department of Electrical and Computer Engineering*
*University of Illinois Chicago*

**Hulya Seferoglu**                                                                        *hulya@uic.edu*
*Department of Electrical and Computer Engineering*
*University of Illinois Chicago*

**Reviewed on OpenReview:** *https://openreview.net/forum?id=F5hgpQ1Ccd*

## Abstract

This paper focuses on reducing the communication cost of federated learning by exploring generalization bounds and representation learning. We first characterize a tighter generalization bound for one-round federated learning based on local clients' generalizations and heterogeneity of data distribution (non-iid scenario). We also characterize a generalization bound in R-round federated learning and its relation to the number of local updates (local stochastic gradient descents (SGDs)). Then, based on our generalization bound analysis and its interpretation through representation learning, we infer that less frequent aggregations for the representation extractor (typically corresponds to initial layers) compared to the head (usually the final layers) leads to the creation of more generalizable models, particularly in non-iid scenarios. We design a novel Federated Learning with Adaptive Local Steps (FedALS) algorithm based on our generalization bound and representation learning analysis. FedALS employs varying aggregation frequencies for different parts of the model, so reduces the communication cost. The paper is followed with experimental results showing the effectiveness of FedALS. Our codes are available for reproducibility.

## 1 Introduction

Federated learning advocates that multiple clients collaboratively train machine learning models under the coordination of a parameter server (central aggregator) (McMahan et al., 2016) by leveraging all clients' computational capacities. Despite its promise, federated learning suffers from high communication costs between clients and the parameter server. In particular, exchanging machine learning models between clients and the parameter server is costly, especially for large models, which are typical in today's machine learning applications (Konecný et al., 2016; Zhang et al., 2013; Barnes et al., 2020; Braverman et al., 2015; Zibaeirad et al., 2024). Furthermore, the uplink bandwidth of clients may be limited, time-varying and expensive. Thus, there is an increasing interest in reducing the communication cost of federated learning via (i) multiple local updates, also known as "Local SGD" (Stich, 2018; Stich & Karimireddy, 2019; Wang & Joshi, 2018), (ii) pruning Wangni et al. (2017); Jiang et al. (2022), (iii) quantization Bernstein et al. (2018); Reisizadeh et al. (2020), etc. In this paper, we take a dramatically different and complementary approach: *Aggregating the different parts of a machine learning model at different frequencies in a federated learning setup.*

The primary purpose of communication in federated learning is to periodically aggregate local models to reduce the consensus distance among clients. This practice helps maintain the overall optimization process on a trajectory toward global optimization. It is important to note that when the consensus distance among clients becomes substantial, the convergence rate reduces. This occurs as individual clients gradually veer towards their respective local optima without being synchronized with the models from other clients. This

issue is amplified when the data distribution among clients is non-iid. It has been demonstrated that the consensus distance is correlated to (i) the randomness in each client's own dataset, which causes variation in consecutive local gradients, as well as (ii) the dissimilarity in loss functions among clients due to non-iidness (Stich & Karimireddy, 2019; Gholami & Seferoglu, 2024). More specifically, the consensus distance at iteration $t$ is defined as $\frac{1}{K} \sum_{k=1}^{K} \|\hat{\boldsymbol{\theta}}_t - \boldsymbol{\theta}_{k,t}\|^2$, where $\hat{\boldsymbol{\theta}}_t = \frac{1}{K} \sum_{k=1}^{K} \boldsymbol{\theta}_{k,t}$, $K$ is the number of clients, $\boldsymbol{\theta}_{k,t}$ is the local model at client $k$ at iteration $t$, and $\|\cdot\|^2$ is squared $l_2$ norm. Note that the consensus distance goes to zero when global aggregation is performed at each communication round. This makes the communication of models between clients and the parameter server crucial, but this introduces significant communication overhead. This paper aims to reduce federated learning communication overhead through the following contributions.

_Contribution I: Improved Generalization Error Bound._ The generalization error of a learning model is defined as the difference between the model's empirical risk and population risks. (We provide a mathematical definition in Section 3). Existing approaches for training models mostly minimize the empirical risk or its variants. However, a small population risk is desired, showing how well the model performs in the test phase as it denotes the loss that occurs when new samples are randomly drawn from the distribution. Note that a small empirical risk and a reduced generalization error correspond to a low population risk. Thus, there is an increasing interest in establishing an upper limit for the generalization error and understanding the underlying factors that affect the generalization error. The generalization error analysis is also important to quantitatively assess the generalization characteristics of trained models, provide reliable guarantees concerning their anticipated performance quality, and design new models and systems.

In this paper, we offer a tighter generalization bound compared to the state of the art Barnes et al. (2022); Yagli et al. (2020); Sun et al. (2023) for federated learning, considering local clients' generalizations and non-iidness (i.e., heterogeneous data distribution across the clients).

_Contribution II: Representation Learning Interpretation._ Recent studies have demonstrated that the concept of representation learning is a promising approach to reducing the communication cost of federated learning (Collins et al., 2021). This is achieved by leveraging the shared representations in all clients' datasets. For example, let us consider a federated learning application for image classification, where different clients have datasets of different animals. Despite each client having a different dataset (one client has dog images, another has cat images, etc.), these images usually have common features such as an eye/ear shape. These shared features, typically extracted in the same way for different types of animals, require consistent layers of a neural network to extract them, whether the animal is a dog or a cat. As a result, these layers demonstrate similarity (i.e., less variation) across clients even when the datasets are non-iid. This implies that the consensus distance for this part of the model (feature extraction) is likely smaller. Based on these observations, our key idea is to reduce the aggregation frequency of the layers that show high similarity, where these layers are updated locally between consecutive aggregations. This approach would reduce the communication cost of federated learning as some layers are aggregated, hence their parameters are exchanged, less frequently. The next example scratches the surface of the problem for a toy example.

**Example 1.** *We consider a federated learning setup of five clients with a central parameter server to train a ResNet-20 (He et al., 2015) on a heterogeneous partition of CIFAR-10 dataset (Krizhevsky, 2009). We use Federated Averaging (FedAvg) (McMahan et al., 2016) as an aggregation algorithm since it is the dominant algorithm in federated learning. We applied FedAvg with 50 local steps prior to each averaging step, denoted as $\tau = 50$. Non-iidness is introduced by allocating 2 classes to each client. Finally, we evaluate the quantity of the average consensus distance for each model layer during the optimization in Fig. 1. It is clear that the initial layers have smaller consensus distance as compared to the final layers. This is due to initial layers' role in extracting representations from input data and their higher similarity across clients.*

The above example indicates that initial layers show higher similarity, so they can be aggregated less frequently. Additionally, several empirical studies (Reddi et al., 2021; Yu et al., 2020) show that federated learning with multiple local updates per round learns a generalizable representation and is unexpectedly successful in non-iid settings. This view is supported by recent theoretical work showing that as you go deeper into a network, the way it represents information changes in a structured way. For example, Li & Sompolinsky (2021) and Zavatone-Veth et al. (2022) have found that a layer's representation is a blend of

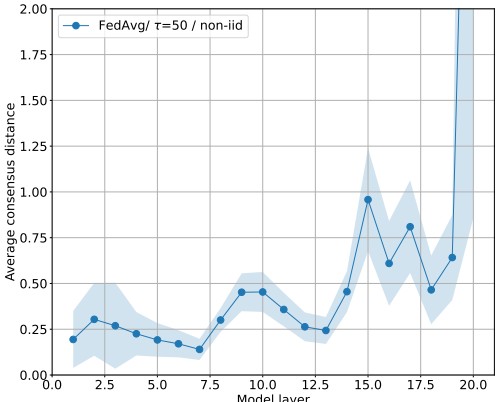

Figure 1: Average consensus distance over time for different layers during ResNet-20 training with FedAvg on CIFAR-10 (5 clients, non-iid, 2 classes per client) shows that early layers have lower consensus distance.

the input data's structure and the learning task's structure. Critically, they showed that the influence of the task becomes stronger in deeper layers, meaning early layers focus more on general input features while later layers adapt to focus on what's needed to solve the task. Also Bordelon & Pehlevan (2022), by doing a study on learning dynamics, confirmed this, showing that deeper layers change their representations more significantly during training to ultimately align with the final goal. Taken together, these papers provide a strong theoretical basis for the idea that networks learn by gradually shifting their focus from representing the input data in early layers to representing the task solution in later layers. These studies encourage us to delve deeper into investigating how local updates and model aggregation frequency for the model's representation extractor and the head affect model generalization.

In this paper, based on our improved generalization bound analysis and our representation learning interpretation of this analysis, we showed for the first time that employing different frequencies of aggregation, *i.e.,* the number of local updates (local SGDs), for the representation extractor (typically corresponding to initial layers) and the head (final label prediction layers), leads to the creation of more generalizable models particularly in non-iid scenarios.

*Contribution III: Design of FedALS.* We design a novel Federated Learning with Adaptive Local Steps (FedALS) algorithm based on our generalization error bound analysis and its representation learning interpretation. FedALS employs varying aggregation frequencies for different parts of the model.

*Contribution IV: Evaluation.* We evaluate the performance of FedALS using deep neural network model ResNet-20 for CIFAR-10, CIFAR-100 (Krizhevsky, 2009), SVHN (Netzer et al., 2011), and MNIST (Lecun et al., 1998) datasets. We also estimate the impact of FedALS on large language models (LLMs) in fine-tuning OPT-125M (Zhang et al., 2022) on the Multi-Genre Natural Language Inference (MultiNLI) corpus (Williams et al., 2018). We consider both iid and non-iid data distributions. Experimental results confirm that FedALS outperforms the baselines in terms of accuracy in non-iid setups while also saving on communication costs across all setups.

## 2 Related Work

There has been increasing interest in distributed learning recently, largely driven by Federated Learning. Several studies have highlighted that these algorithms achieve convergence to a global optimum or a stationary point of the overall objective, particularly in convex or non-convex scenarios (Stich & Karimireddy, 2019; Stich, 2018; Gholami & Seferoglu, 2024; Lian et al., 2018; Kairouz et al., 2019). However, it is widely accepted that communication cost is the major bottleneck for these techniques in large-scale optimization applications (Konecný et al., 2016; Lin et al., 2017). To tackle this issue, two primary strategies are put forth: the utilization of mini-batch parallel SGD, and the adoption of Local SGD. These approaches aim to enhance the equilibrium between computation and communication. Woodworth et al. (2020b;a) attempt to theoretically capture the distinction to comprehend under what circumstances Local SGD outperforms minibatch SGD.

Local SGD appears to be more intuitive compared to minibatch SGD, as it ensures progress towards the optimum even in cases where workers are not communicating and employing a mini-batch size that is too large may lead to a decrease in performance (Lin et al., 2017). However, due to the fact that individual gradients for each worker are computed at distinct instances, this technique brings about residual errors. As a result, a compromise arises between reducing communication rounds and introducing supplementary errors into the gradient estimations. This becomes increasingly significant when data is unevenly distributed across nodes. There are several decentralized algorithms that have been shown to mitigate heterogeneity (Karimireddy et al., 2019; Liu et al., 2023). One prominent example is the Stochastic Controlled Averaging algorithm (SCAFFOLD) (Karimireddy et al., 2019), which addresses the node drift caused by non-iid characteristics of data distribution. They establish the notion that SCAFFOLD demonstrates a convergence rate at least equivalent to SGD, ensuring convergence even when dealing with highly non-iid datasets.

However, despite these factors, multiple investigations (Reddi et al., 2021; Yu et al., 2020; Lin et al., 2020; Gu et al., 2023), have noted that the model trained using FedAvg and incorporating multiple Local SGD per round exhibits unexpected effectiveness when subsequently fine-tuned for individual clients in non-iid Federated learning setting. This implies that the utilization of FedAvg with several local updates proves effective in acquiring a valuable data representation, which can later be employed on each node for downstream tasks. Following this line of reasoning, our justification will be based on the argument that the Local SGD component of FedAvg contributes to improving performance in heterogeneous scenarios by facilitating the acquisition of models with enhanced generalizability.

An essential characteristic of machine learning systems is their capacity to extend their performance to novel and unseen data. This capacity, referred to as generalization, can be expressed within the framework of statistical learning theory. There has been a line of research to characterize generalization bound in FL Wang & Ma (2023); Mohri et al. (2019). More recently Barnes et al. (2022); Sun et al. (2023); Yagli et al. (2020); Sefidgaran et al. (2024) considered this problem and gave upper bounds on the expected generalization error for FL in iid setting in terms of the local generalizations of clients. These studies demonstrate a $\frac{1}{K}$ dependency on the number of nodes. Motivated by this, our research focuses on analyzing generalization that improves the dependency to $\frac{1}{K^2}$ in iid setting. We then use the derived insights to introduce FedALS, aiming to enhance generalization in non-iid setup.

Our work differentiates itself from personalized federated learning work Collins et al. (2021) in a way that our work focuses on a general federated learning setup, where a global model is trained collectively while personalized federated learning advocates training a model from each client's perspective. Thus, the results are strikingly different; They show that the heads should be trained locally, while our work FedALS shows that heads should be aggregated more frequently. They also consider only linear models, while our work is generic. Collins et al. (2022) analyze the behavior of FedAvg in multi-task linear regression with a common representation, focusing on the convergence behavior of the algorithm. They prove a faster convergence rate when more than one local step is deployed per round. However, they do not address generalization in the context of statistical learning, which is a crucial aspect of our work.

## 3 Background and Problem Statement

### 3.1 Preliminaries and Notation

We consider that we have $K$ clients/nodes in our system, and each node has its own portion of the dataset. For example, node $k$ has a local dataset $\boldsymbol{S}_k = \{\boldsymbol{z}_{k,1}, ..., \boldsymbol{z}_{k,n_k}\}$, where $\boldsymbol{z}_{k,i} = (\boldsymbol{x}_{k,i}, \boldsymbol{y}_{k,i})$ is drawn from a distribution $\mathcal{D}_k$ over $\mathcal{X} \times \mathcal{Y}$, where $\mathcal{X}$ is the input space and $\mathcal{Y}$ is the label space. We consider $\mathcal{X} \subseteq \mathbb{R}^d$ and $\mathcal{Y} \subseteq \mathbb{R}$. The size of the local dataset at node $k$ is $n_k$. The dataset across all nodes is defined as $\boldsymbol{S} = \{\boldsymbol{S}_1, ..., \boldsymbol{S}_K\}$. Data distribution across the nodes could be independent and identically distributed (iid) or non-iid. In iid setting, we assume that $\mathcal{D}_1 = ... = \mathcal{D}_K = \mathcal{D}$ holds. On the other hand, non-iid setting covers all cases where this equality does not hold.

We assume that $M_{\boldsymbol{\theta}} = \mathcal{A}(\boldsymbol{S})$ represents the output of a possibly stochastic function denoted as $\mathcal{A}(\boldsymbol{S})$, where $M_{\boldsymbol{\theta}} : \mathcal{X} \to \mathcal{Y}$ represents the learned model parameterized by $\boldsymbol{\theta}$. We consider a real-valued loss function denoted as $l(M_{\boldsymbol{\theta}}, \boldsymbol{z})$, which assesses the model $M_{\boldsymbol{\theta}}$ based on a sample $\boldsymbol{z}$.

### 3.2 Generalization Error

We first define an empirical risk on dataset $\boldsymbol{S}$ as

$$R_{\boldsymbol{S}}(M_{\boldsymbol{\theta}}) = \mathbb{E}_{k \sim \mathcal{K}} R_{\boldsymbol{S}_k}(M_{\boldsymbol{\theta}}) = \mathbb{E}_{k \sim \mathcal{K}} \frac{1}{n_k} \sum_{i=1}^{n_k} l(M_{\boldsymbol{\theta}}, \boldsymbol{z}_{k,i}), \tag{1}$$

where $\mathcal{K}$ is an arbitrary distribution over nodes to weight different local risk contributions in the global risk. Specifically, $\mathcal{K}(k)$ represents the contribution of node $k$'s loss in the global loss. In the most conventional case, it is usually assumed to be uniform across all nodes, i.e., $\mathcal{K}(k) = \frac{1}{K}$ for all $k$. $R_{\boldsymbol{S}_k}(M_{\boldsymbol{\theta}})$ is the empirical risk for model $M_{\boldsymbol{\theta}}$ on local dataset $\boldsymbol{S}_k$. We further define a population risk for model $M_{\boldsymbol{\theta}}$ as

$$R(M_{\boldsymbol{\theta}}) = \mathbb{E}_{k \sim \mathcal{K}} R_k(M_{\boldsymbol{\theta}}) = \mathbb{E}_{k \sim \mathcal{K}, \boldsymbol{z} \sim \mathcal{D}_k} l(M_{\boldsymbol{\theta}}, \boldsymbol{z}), \tag{2}$$

where $R_k(M_{\boldsymbol{\theta}})$ is the population risk on node $k$'s data distribution.

Now, we can define the generalization error for dataset $\boldsymbol{S}$ and function $\mathcal{A}(\boldsymbol{S})$ as

$$\Delta_{\mathcal{A}}(\boldsymbol{S}) = R(\mathcal{A}(\boldsymbol{S})) - R_{\boldsymbol{S}}(\mathcal{A}(\boldsymbol{S})). \tag{3}$$

The expected generalization error is expressed as $\mathbb{E}_{\boldsymbol{S}} \Delta_{\mathcal{A}}(\boldsymbol{S})$, where $\mathbb{E}_{\boldsymbol{S}}[\cdot] = \mathbb{E}_{\{\boldsymbol{S}_k \sim \mathcal{D}_k^{n_k}\}_{k=1}^{K}}[\cdot]$ is used for the sake of notation convenience.

### 3.3 Federated Learning

We consider a federated learning scenario with $K$ nodes/clients and a centralized parameter server. The nodes update their localized models to minimize their empirical risk $R_{\boldsymbol{S}_k}(M_{\boldsymbol{\theta}})$ on local dataset $\boldsymbol{S}_k$, while the parameter server aggregates the local models to minimize the empirical risk $R_{\boldsymbol{S}}(M_{\boldsymbol{\theta}})$. Due to connectivity and privacy constraints, the clients do not exchange their data with each other. One of the most widely used federated learning algorithms is FedAvg (McMahan et al., 2016), which we explain in detail next.

At round $r$ of FedAvg, each node $k$ trains its model $M_{\boldsymbol{\theta}_{k,r}} = \mathcal{A}_{k,r}(\boldsymbol{S}_k)$ locally using the function/algorithm $\mathcal{A}_{k,r}$. The local models $M_{\boldsymbol{\theta}_{k,r}}$ are transmitted to the central parameter server, which merges the received local models to aggregated model parameters $\hat{\boldsymbol{\theta}}_{r+1} = \hat{\mathcal{A}}(\boldsymbol{\theta}_{1,r}, ..., \boldsymbol{\theta}_{K,r})$, where $\hat{\mathcal{A}}$ is the aggregation function. In FedAvg, the aggregation function calculates an average, so the aggregated model is expressed as

$$\hat{\boldsymbol{\theta}}_{r+1} = \mathbb{E}_{k \sim \mathcal{K}} \boldsymbol{\theta}_{k,r}. \tag{4}$$

Subsequently, the aggregated model is transmitted to all nodes. This process continues for $R$ rounds. The final model after $R$ rounds of FedAvg is $\mathcal{A}(S)$.

The local models are usually trained using stochastic gradient descent (SGD) at each node. To reduce the communication cost needed between the nodes and the parameter server, each node executes multiple SGD steps using its local data after receiving an aggregated model from the parameter server. To be precise, we have the aggregated model parameters at round $r$ as $\hat{\boldsymbol{\theta}}_r$. Specifically, upon receiving $\hat{\boldsymbol{\theta}}_r$, node $k$ computes

$$\boldsymbol{\theta}_{k,r,t+1} = \boldsymbol{\theta}_{k,r,t} - \frac{\eta}{|\mathcal{B}_{k,r,t}|} \sum_{i \in \mathcal{B}_{k,r,t}} \nabla l(M_{\boldsymbol{\theta}_{k,r,t}}, \boldsymbol{z}_{k,i}) \tag{5}$$

for $t = 0, \ldots, \tau - 1$, where $\tau$ is the number of local SGD steps, $\boldsymbol{\theta}_{k,r,0}$ is defined as $\boldsymbol{\theta}_{k,r,0} = \hat{\boldsymbol{\theta}}_r$, $\eta$ is the learning rate, $\mathcal{B}_{k,r,t}$ is the batch of samples used in local step $t$ of round $r$ in node $k$, $\nabla$ is the gradient, and $|\cdot|$ shows the size of a set. Upon completing the local steps in round $r$, each node transmits $\boldsymbol{\theta}_{k,r} = \boldsymbol{\theta}_{k,r,\tau}$ to the parameter server to calculate $\hat{\boldsymbol{\theta}}_{r+1}$ as in (4).

### 3.4 Representation Learning

Our approach for analyzing the generalization error bounds for federated learning, specifically focusing on FedAvg, uses representation learning, which we explain next.

We consider a class of models that consist of a representation extractor (*e.g.,* ResNet). Let $\boldsymbol{\theta}$ be the model $M_{\boldsymbol{\theta}}$'s parameters. We can decompose $\boldsymbol{\theta}$ into two sets: $\boldsymbol{\phi}$ containing the representation extractor's parameters and $\boldsymbol{h}$ containing the head parameters, *i.e.,* $\boldsymbol{\theta} = [\boldsymbol{\phi}, \boldsymbol{h}]$. $M_{\boldsymbol{\phi}}$ is a function that maps from the original input space to some feature space, *i.e.,* $M_{\boldsymbol{\phi}} : \mathbb{R}^d \to \mathbb{R}^{d'}$, where usually $d' \ll d$. The function $M_{\boldsymbol{h}}$ performs a low complexity mapping from the representation space to the label space, which can be expressed as $M_{\boldsymbol{h}} : \mathbb{R}^{d'} \to \mathbb{R}$.

For any $\boldsymbol{x} \in \mathcal{X}$, the output of the model is $M_{\boldsymbol{\theta}}(\boldsymbol{x}) = (M_{\boldsymbol{h}} \circ M_{\boldsymbol{\phi}})(\boldsymbol{x}) = M_{\boldsymbol{h}}(M_{\boldsymbol{\phi}}(\boldsymbol{x}))$. For instance, if $M_{\boldsymbol{\theta}}$ is a neural network, $M_{\boldsymbol{\phi}}$ represents several initial layers of the network, which are typically designed to extract meaningful representations from the neural network's input. On the other hand, $M_{\boldsymbol{h}}$ denotes the final few layers that lead to the network's output.

## 4  Improved Generalization Bounds

In this section, we derive generalization bounds for FedAvg based on clients' local generalization performances in a general non-iid setting for the first time in the literature. First, we start with one-round FedAvg and analyze its generalization bound. Then, we extend our analysis to $R-$round FedAvg.

### 4.1  One-Round Generalization Bound

In the following theorem, we determine the generalization bound for one round of FedAvg.

**Theorem 4.1.** *Let $l(M_{\boldsymbol{\theta}}, \boldsymbol{z})$ be $\mu$-strongly convex and $L$-smooth in $M_{\boldsymbol{\theta}}$. $M_{\boldsymbol{\theta}_k} = \mathcal{A}_k(\boldsymbol{S}_k)$ represents the model obtained from Empirical Risk Minimization (ERM) algorithm on local dataset $\boldsymbol{S}_k$, i.e., $M_{\boldsymbol{\theta}_k} = \arg\min_M \sum_{i=1}^{n_k} l(M, \boldsymbol{z}_{k,i})$, and $M_{\hat{\boldsymbol{\theta}}} = \mathcal{A}(\boldsymbol{S})$ is the model after one round of FedAvg ($\hat{\boldsymbol{\theta}} = \mathbb{E}_{k \sim \mathcal{K}} \boldsymbol{\theta}_k$). Then, the expected generalization error, $\mathbb{E}_{\boldsymbol{S}} \Delta_{\mathcal{A}}(\boldsymbol{S})$, is upper bounded with*

$$
\mathbb{E}_{k \sim \mathcal{K}} \left[ \frac{L\mathcal{K}(k)^2}{\mu} \underbrace{\mathbb{E}_{\boldsymbol{S}_k} \Delta_{\mathcal{A}_k}(\boldsymbol{S}_k)}_{\textit{Expected local generalization}} + 2\sqrt{\frac{L}{\mu}} \mathcal{K}(k) \bigg( \underbrace{\mathbb{E}_{\boldsymbol{S}} \delta_{k,\mathcal{A}}(\boldsymbol{S})}_{\textit{Expected non-iidness}} \underbrace{\mathbb{E}_{\boldsymbol{S}_k} \Delta_{\mathcal{A}_k}(\boldsymbol{S}_k)}_{\textit{Expected local generalization}} \bigg)^{\frac{1}{2}} \right], \quad (6)
$$

*where $\delta_{k,\mathcal{A}}(\boldsymbol{S}) = R_{\boldsymbol{S}_k}(\mathcal{A}(\boldsymbol{S})) - R_{\boldsymbol{S}_k}(\mathcal{A}_k(\boldsymbol{S}_k))$ shows the level of non-iidness at client $k$ for function $\mathcal{A}$ on dataset $\boldsymbol{S}$.*

*Proof:* The proof of Theorem 4.1 is in Appendix B. □

*Remark* 4.2. We note that this theorem and its proof assume that all clients participate in learning. The other scenario is that not all clients participate in the learning procedure. We can consider the following two cases when not all clients participate in the learning procedure.

*Case I:* Sampling $\hat{K}$ clients with replacement based on distribution $\mathcal{K}$, followed by averaging the local models with equal weights.

*Case II:* Sampling $\hat{K}$ clients without replacement uniformly at random, then performing weighted averaging of local models. Here, the weight of client $k$ is rescaled to $\frac{\mathcal{K}(k)K}{\hat{K}}$.

The generalization error results in these cases are affected by substituting $\frac{1}{\hat{K}}$ and $\frac{\mathcal{K}(k)K}{\hat{K}}$ instead of $\mathcal{K}(k)$ in (6) for cases I and II, respectively. The detailed proof is provided in Appendix D.

**Discussion.** Note that there are two terms in the generalization error bound: (i) local generalization of each client that shows more generalizable local models lead to a better generalization of the aggregated model, (ii) non-iidness of each client which deteriorates generalization. Theorem 4.1 reveals a factor of $\mathcal{K}(k)^2$ for the first term, which is the sole term in the iid setting. For example, in the uniform case ($\mathcal{K}(k) = \frac{1}{K}$), we will observe an improvement with a factor of $\frac{1}{K^2}$ for the iid case. This represents an enhancement compared to the state of the art (Barnes et al., 2022; Sun et al., 2023; Yagli et al., 2020), which only demonstrates a factor of $\frac{1}{K}$. As a result, after the averaging process carried out by the central parameter server, the generalization error is reduced by a factor of $\frac{1}{K^2}$ in iid case. This is interesting because one would normally expect an improvement

of $\frac{1}{K}$ based on the linear increase in the collective dataset, but this shows an additional improvement of $\frac{1}{K}$ as well.

On the other hand, we do not see a similar behavior in non-iid case. In other words, the expected generalization error bound does not necessarily decrease with averaging. These results show why FedAvg works well in iid setup but not necessarily in non-iid setup. This observation motivates us to design a new federated learning approach for non-iid setup. The question in this context is what the new federated learning design should be. To answer this question, we analyze $R-$round generalization bound in the next section.

### 4.2 $R-$Round Generalization Bound

In this setup, after $R$ rounds, there is a sequence of weights $\{\hat{\boldsymbol{\theta}}_r\}_{r=1}^R$ and the final model is $\hat{\boldsymbol{\theta}}_R$. We consider that at round $r$, each node constructs its updated model as in (5) by taking $\tau$ gradient steps starting from $\hat{\boldsymbol{\theta}}_r$ with respect to $\tau$ random mini-batches $Z_{k,r} = \bigcup \{\mathcal{B}_{k,r,t}\}_{t=0}^{\tau-1}$ drawn from the local dataset $\boldsymbol{S}_k$. For this type of iterative algorithm, we consider the following averaged empirical risk

$$\frac{1}{R}\sum_{r=1}^R \mathbb{E}_{k\sim\mathcal{K}}\left[\frac{1}{|Z_{k,r}|}\sum_{i\in Z_{k,r}} l(M_{\hat{\boldsymbol{\theta}}_r}, \boldsymbol{z}_{k,i})\right]. \tag{7}$$

The corresponding generalization error, $\Delta_{FedAvg}(\boldsymbol{S})$, is

$$\frac{1}{R}\sum_{r=1}^R \mathbb{E}_{k\sim\mathcal{K}}\left[\mathbb{E}_{\boldsymbol{z}\sim\mathcal{D}_k} l(M_{\hat{\boldsymbol{\theta}}_r}, \boldsymbol{z}) - \frac{1}{|Z_{k,r}|}\sum_{i\in Z_{k,r}} l(M_{\hat{\boldsymbol{\theta}}_r}, \boldsymbol{z}_{k,i})\right]. \tag{8}$$

Note that the expression in (8) differs slightly from the end-to-end generalization error that would be obtained by considering the final model $M_{\hat{\boldsymbol{\theta}}_R}$ and the entire dataset $\boldsymbol{S}$. More specifically, (8) is an average of the generalization errors measured at each round, similar to Barnes et al. (2022)). We anticipate that the generalization error diminishes with the increasing number of data samples, so this generalization error definition yields a more cautious upper limit and serves as a sensible measure. The next theorem characterizes the expected generalization error bounds for $R-$Round FedAvg in iid and non-iid settings.

**Theorem 4.3.** *Let $l(M_{\boldsymbol{\theta}}, \boldsymbol{z})$ be $\mu$-strongly convex and $L$-smooth in $M_{\boldsymbol{\theta}}$. Local models at round $r$ are calculated by doing $\tau$ local gradient descent steps (5), and the local gradient variance is bounded by $\sigma^2$, i.e., $\mathbb{E}_{\boldsymbol{z}\sim\mathcal{D}_k}\|\nabla l(M_{\boldsymbol{\theta}}, \boldsymbol{z}) - \mathbb{E}_{\boldsymbol{z}\sim\mathcal{D}_k}\nabla l(M_{\boldsymbol{\theta}}, \boldsymbol{z})\|^2 \leq \sigma^2$. The aggregated model at round $r$, $M_{\hat{\boldsymbol{\theta}}_r}$, is obtained by performing FedAvg, and the data points used in round $r$ (i.e., $Z_{k,r}$) are sampled without replacement. Then the average generalization error, $\mathbb{E}_{\boldsymbol{S}} \Delta_{FedAvg}(\boldsymbol{S})$, is upper bounded by*

$$\frac{1}{R}\sum_{r=1}^R \mathbb{E}_{k\sim\mathcal{K}}\left[\frac{2L\mathcal{K}(k)^2}{\mu}A + \sqrt{\frac{8L}{\mu}}\mathcal{K}(k)(AB)^{\frac{1}{2}}\right], \tag{9}$$

*where,*

$$A = \tilde{O}\left(\sqrt{\frac{\mathcal{C}(M_{\boldsymbol{\theta}})}{|Z_{k,r}|}} + \frac{\sigma^2}{\mu\tau}\right),$$

$$B = \tilde{O}\left(\mathbb{E}_{\{Z_{k,r}\}_{k=1}^K} \delta_{k,\mathcal{A}}(\{Z_{k,r}\}_{k=1}^K) + \frac{\sigma^2}{\mu\tau}\right).$$

*$\tilde{O}$ hides constants and poly-logarithmic factors, and $\mathcal{C}(M_{\boldsymbol{\theta}})$ shows the complexity of the model class of $M_{\boldsymbol{\theta}}$.*

*Proof:* The proof of Theorem 4.3 is in Appendix C. □

The generalization error bound in (9) depends on the following parameters: (i) number of rounds; $R$, (ii) number of samples used in every round; $|Z_{k,r}|$, (iii) the complexity of the model class; $\mathcal{C}(M_{\boldsymbol{\theta}})$, non-iidness; $\delta_{k,\mathcal{A}}(\{Z_{k,r}\}_{k=1}^K)$, number of local steps in each round; $\tau$. We note that (9) also depends on $K$ (more specifically $\mathcal{K}$), but this dependence is similar to the discussion we had for one-round generalization, so we skip it here.

---

**Algorithm 1** FedALS

---

**Input**: Initial model $\{\boldsymbol{\theta}_{k,0,0} = [\boldsymbol{\phi}_{k,0,0}, \boldsymbol{h}_{k,0,0}]\}_{k=1}^K$, learning rate $\eta$, number of local steps for the head $\tau$, adaptation coefficient $\alpha$.

  1: **for** Round $r$ in $0, ..., R-1$ **do**
  2:    **for** Node $k$ in $1, ..., K$ **in parallel do**
  3:       **for** Local step $t$ in $0, ..., \tau-1$ **do**
  4:          Sample the batch $\mathcal{B}_{k,r,t}$ from $\mathcal{D}_k$.
  5:          $\boldsymbol{\theta}_{k,r,t+1} = \boldsymbol{\theta}_{k,r,t} - \frac{\eta}{|\mathcal{B}_{k,r,t}|} \sum_{i \in \mathcal{B}_{k,r,t}} \nabla l(M_{\boldsymbol{\theta}_{k,r,t}}, \boldsymbol{z}_{k,i})$
  6:          **if** $\mod(r\tau+t, \tau) = 0$ **then**
  7:             Send $\boldsymbol{h}_{k,r,t}$ to the server.
  8:             Receive the aggregated head: $\boldsymbol{h}_{k,r,t} \leftarrow \frac{1}{K} \sum_{k=1}^K \boldsymbol{h}_{k,r,t}$
  9:          **end if**
10:         **if** $\mod(r\tau+t, \alpha\tau) = 0$ **then**
11:           Send $\boldsymbol{\phi}_{k,r,t}$ to the server.
12:           Receive the aggregated Representation extractor: $\boldsymbol{\phi}_{k,r,t} \leftarrow \frac{1}{K} \sum_{k=1}^K \boldsymbol{\phi}_{k,r,t}$
13:         **end if**
14:       **end for**
15:       $\boldsymbol{\theta}_{k,r+1,0} = \boldsymbol{\theta}_{k,r,\tau}$
16:    **end for**
17: **end for**
18: **return** $\hat{\boldsymbol{\theta}}_R = \frac{1}{K} \sum_{k=1}^K \boldsymbol{\theta}_{k,R,0}$

---

## 5   FedALS: Federated Learning with Adaptive Local Steps

The number of samples and local steps in each round are crucial for minimizing the generalization error bound, particularly in non-iid scenarios, as described in (9), where the error bound is looser compared to the iid setup. Our key insight in this paper is that reducing the aggregation frequency, which increases both $\tau$ and $|Z_{k,r}|$, can lead to a smaller generalization error bound according to (9).

Increasing $\tau$ also increases the non-iidness among nodes, defined as:

$$\delta_{k,\mathcal{A}}(\{Z_{k,r}\}_{k=1}^K) = R_{Z_{k,r}}(M_{\boldsymbol{h}} \circ M_{\boldsymbol{\phi}}) - R_{Z_{k,r}}(M_{\boldsymbol{h}_k} \circ M_{\boldsymbol{\phi}_k}),$$

where $M_{\boldsymbol{h}} \circ M_{\boldsymbol{\phi}}$ represents the aggregated model $\mathcal{A}(\{Z_{k,r}\}_{k=1}^K)$, and $M_{\boldsymbol{h}_k} \circ M_{\boldsymbol{\phi}_k}$ represents the local model $\mathcal{A}_k(Z_{k,r})$. This happens because the model moves closer to the local optimum of node $k$'s loss, thereby reducing $R_{Z_{k,r}}(M_{\boldsymbol{h}_k} \circ M_{\boldsymbol{\phi}_k})$.

In the context of representation learning, as discussed in the Introduction, the representation extractors $M_{\boldsymbol{\phi}}$ and $M_{\boldsymbol{\phi}_k}$ are much more similar compared to the task-specific heads $M_{\boldsymbol{h}}$ and $M_{\boldsymbol{h}_k}$. Therefore, increasing the number of local steps for the representation extractor does not significantly increase non-iidness, unlike increasing local steps for the head.

The main idea behind FedALS is to improve generalization of the model by increasing the number of local steps while maintaining a low level of non-iidness. This can be achieved if $\tau_{M_{\boldsymbol{\phi}}}$ is set larger than $\tau_{M_{\boldsymbol{h}}}$, where $\tau_{M_{\boldsymbol{\phi}}}$ and $\tau_{M_{\boldsymbol{h}}}$ are the corresponding number of local iterations for $M_{\boldsymbol{\phi}}$ and $M_{\boldsymbol{h}}$, respectively. FedALS in Algorithm 1 divides the model into two parts: (i) the representation extractor, denoted as $M_{\boldsymbol{\phi}}$, and (ii) the head, denoted as $M_{\boldsymbol{h}}$. Additionally, we introduce the parameter $\alpha = \frac{\tau_{M_{\boldsymbol{\phi}}}}{\tau_{M_{\boldsymbol{h}}}}$ as an adaptation coefficient, which can be regarded as a hyperparameter for estimating the true ratio.

## 6   Experimental Results

In this section, we assess the performance of FedALS using ResNet-20 as a deep neural network architecture and OPT-125M as a large language model. We treat the convolutional layers of ResNet-20 as the representation extractor and the final dense layers as the model head. For OPT-125M, we consider the first 10

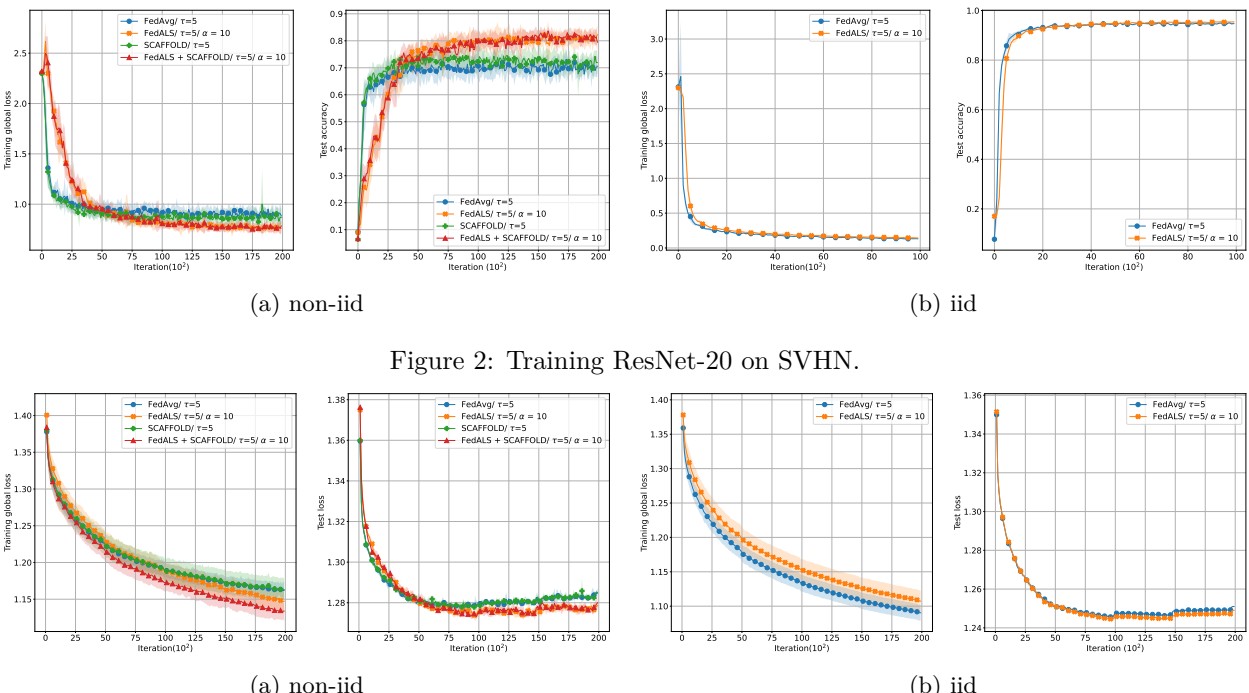

Figure 2: Training ResNet-20 on SVHN.

Figure 3: Fine-tuning OPT-125M on MultiNLI.

layers of the model as the representation extractor. We used the datasets CIFAR-10, CIFAR-100, SVHN, and MNIST for image classification and the Multi-Genre Natural Language Inference (MultiNLI) corpus for the LLM. The experimentation was conducted on a network consisting of five nodes alongside a central server. For image classification, we utilized a batch size of 64 per node. SGD with momentum was employed as the optimizer, with the momentum set to 0.9, and the weight decay to $10^{-4}$. For the LLM fine-tuning, we employed a batch size of 16 sentences from the corpus, and the optimizer used was AdamW. In all the experiments, to perform a grid search for the learning rate, we conducted each experiment by multiplying and dividing the learning rate by powers of two, stopping each experiment after reaching a local optimum learning rate. We repeat each experiment 20 times and present the error bars associated with the randomness of the optimization. In every figure, we include the average and standard deviation error bars. Detailed experimental setup is provided in Appendix E of the supplementary materials.

## 6.1 FedALS in non-iid Setting

In this section, we allocate the dataset to nodes using a non-iid approach. For image classification, we initially sorted the data based on their labels and subsequently divided it among nodes following this sorted sequence. In MultiNLI, we sorted the sentences based on their genre.

In this scenario, we can observe in Fig. 2a, 3a the anticipated performance improvement through the incorporation of different local steps across the model. By utilizing parameters $\tau = 5$ and $\alpha = 10$ in FedALS, it becomes apparent that aggregation and communication costs are reduced compared to FedAvg with the same $\tau$ value of 5. This implies that the initial layers perform aggregation at every 50 iterations. This reduction in the number of communications is accompanied by enhanced model generalization stemming from the larger number of local steps in the initial layers, which contributes to an overall performance enhancement. Thus, our approach in FedALS is beneficial for both communication efficiency and enhancing model generalization performance simultaneously.

## 6.2 FedALS in iid Setting

The results for the iid setting are presented in Fig. 2b, 3b. In order to obtain these results, the data is shuffled, and then evenly divided among nodes. We note that in this situation, the performance improvement

Table 1: Test Performance for 5 nodes FL; Accuracy after training ResNet-20 and test loss after fine-tuning OPT-125M in iid and non-iid settings with $\tau = 5$ and $\alpha = 10$.

| Model/Dataset | FedAvg | | FedALS | | SCAFFOLD | FedALS + SCAFFOLD |
|---|---|---|---|---|---|---|
| | IID | Non-IID | IID | Non-IID | Non-IID | Non-IID |
| ResNet-20/SVHN | $0.948 \pm 0.002$ | $\mathbf{0.701 \pm 0.033}$ | $0.954 \pm 0.002$ | $\mathbf{0.812 \pm 0.021}$ | $0.718 \pm 0.040$ | $0.810 \pm 0.027$ |
| ResNet-20/CIFAR-10 | $0.876 \pm 0.009$ | $\mathbf{0.465 \pm 0.007}$ | $0.887 \pm 0.002$ | $\mathbf{0.522 \pm 0.037}$ | $0.454 \pm 0.071$ | $0.512 \pm 0.010$ |
| ResNet-20/CIFAR-100 | $0.600 \pm 0.007$ | $\mathbf{0.418 \pm 0.014}$ | $0.612 \pm 0.005$ | $\mathbf{0.486 \pm 0.022}$ | $0.412 \pm 0.018$ | $0.482 \pm 0.013$ |
| ResNet-20/MNIST | $0.991 \pm 0.000$ | $\mathbf{0.797 \pm 0.064}$ | $0.991 \pm 0.000$ | $\mathbf{0.821 \pm 0.036}$ | $0.812 \pm 0.055$ | $0.783 \pm 0.118$ |
| OPT-125M/MultiNLI | $1.250 \pm 0.000$ | $\mathbf{1.284 \pm 0.002}$ | $1.248 \pm 0.001$ | $\mathbf{1.277 \pm 0.002}$ | $1.284 \pm 0.001$ | $1.278 \pm 0.002$ |

of FedALS is negligible. This is expected since there is a factor of $\frac{1}{K^2}$ in the generalization in this case, ensuring that we will have nearly the same population risk as the empirical risk.

### 6.3 Compared to and Complementing SCAFFOLD

Karimireddy et al. (2019) introduced an innovative technique called SCAFFOLD, which employs some control variables for variance reduction to address the issue of "client-drift" in local updates. This drift happens when data is heterogeneous (non-iid), causing individual nodes/clients to converge towards their local optima rather than the global optima. While this approach is a significant theoretical advancement in achieving independence from loss function disparities among nodes, it hinges on the assumption of smoothness in the loss functions, which might not hold true for practical deep learning problems in the real world. Additionally, since SCAFFOLD requires the transmission of control variables to the central server, which is of the same size as the models themselves, it results in approximately twice the communication overhead when compared to FedAvg.

Let us consider Fig. 2a, 3a to notice that in real-world deep learning situations, FedALS enhances performance significantly, while SCAFFOLD exhibits slight improvements in specific scenarios. Moreover, we integrated FedALS and SCAFFOLD to concurrently leverage both approaches. The results of the test accuracy in different datasets are summarized in Table 1.

### 6.4 Complementing SCAFFOLD

Karimireddy et al. (2019) introduced an innovative technique called SCAFFOLD, which employs some control variables for variance reduction to address the issue of "client-drift" in local updates. This drift happens when data is heterogeneous (non-iid), causing individual nodes/clients to converge towards their local optima rather than the global optima. Methods like SCAFFOLD are complementary to FedALS. They can be applied concurrently to gain the dual benefits of improved communication efficiency from FedALS while further combating the effects of non-iid data. SCAFFOLD incurs approximately double the communication overhead of FedAvg because it requires sending control variables that are equal in size to the model parameters. However, this overhead can be mitigated by applying FedALS, which improves SCAFFOLD's communication efficiency.

Let us consider Fig. 2a, 3a to notice that in real-world deep learning situations, FedALS enhances performance significantly, while SCAFFOLD exhibits slight improvements in specific scenarios. Moreover, we integrated FedALS and SCAFFOLD to concurrently leverage both approaches. The results of the test accuracy in different datasets are summarized in Table 1.

### 6.5 The Role of $\alpha$ and Communication Overhead

As shown in Table 2, it becomes evident that when we increase $\alpha$ from 1 (FedAvg), we initially witness an enhancement in accuracy owing to improved generalization. However, beyond a certain threshold ($\alpha = 10$), further increment in $\alpha$ ceases to contribute to performance improvement. This is due to the adverse impact of a high number of local steps on the non-iidness. The trade-off we discussed in the earlier sections is evident in this context. We have also demonstrated the impact of FedALS on the communication overhead in this table.

Table 2: The accuracy and communication overhead per client after training ResNet-20 in non-iid setting with $\tau = 5$ and variable $\alpha$.

| VALUE OF $\boldsymbol{\alpha}$ | DATASET | | # OF COMMUNICATED |
|:---:|:---:|:---:|:---:|
| | SVHN | CIFAR-10 | PARAMETERS |
| 1 | $0.7010 \pm 0.0330$ | $0.4651 \pm 0.0071$ | $2.344B$ |
| 5 | $0.8107 \pm 0.0278$ | $0.5201 \pm 0.0302$ | $0.473B$ |
| 10 | $\mathbf{0.8117 \pm 0.0214}$ | $\mathbf{0.5224 \pm 0.0365}$ | $0.239B$ |
| 25 | $0.7201 \pm 0.0549$ | $0.3814 \pm 0.0641$ | $0.099B$ |
| 50 | $0.6377 \pm 0.0520$ | $0.2853 \pm 0.0641$ | $0.052B$ |
| 100 | $0.5837 \pm 0.0715$ | $0.2817 \pm 0.032$ | $0.029B$ |

Table 3: Different Combinations of $\boldsymbol{\phi}, \boldsymbol{h}$ for training ResNet-20 in non-iid setting with $\tau = 5$, $\alpha = 10$.

| VALUE OF $\boldsymbol{L}$ | DATASET | | |
|:---:|:---:|:---:|:---:|
| | SVHN | CIFAR-10 | CIFAR-100 |
| 20 | $0.6991 \pm 0.0160$ | $0.4383 \pm 0.0423$ | $0.4781 \pm 0.0123$ |
| 16 | $\mathbf{0.7112 \pm 0.0471}$ | $\mathbf{0.4687 \pm 0.0111}$ | $\mathbf{0.4782 \pm 0.0087}$ |
| 12 | $0.6760 \pm 0.0474$ | $0.4125 \pm 0.0283$ | $0.4249 \pm 0.0143$ |
| 8 | $0.6381 \pm 0.0428$ | $0.3779 \pm 0.03451$ | $0.4085 \pm 0.0094$ |
| 4 | $0.6339 \pm 0.0446$ | $0.3730 \pm 0.0310$ | $0.4183 \pm 0.0108$ |
| 1 | $0.6058 \pm 0.0197$ | $0.4013 \pm 0.0308$ | $0.3880 \pm 0.0305$ |

### 6.6 Different Combinations of $\phi, h$

In Table 3, we have presented the results of our experiments, illustrating how different combinations of $\phi$ and $h$ influence the model performance in FedALS. The parameter $L$ indicates the number of layers in the model considered as the representation extractor $(\phi)$, while the remaining layers are considered as $h$. We observe that for ResNet-20, choosing $\phi$ to be the first 16 layers and performing less aggregation for them seems to be the most effective option.

## 7 Conclusion

In this paper, we characterized the generalization error bound for one- and R-round federated learning, showing that the one-round bound is tighter than the current state of the art. Our analysis, combined with a representation learning perspective, revealed that less frequent aggregations, resulting in more local updates for the initial layers, lead to more generalizable models, especially in non-iid scenarios. This insight inspired the development of the FedALS algorithm, which increases local steps for the initial layers while performing more averaging for the final layers. Experimental results demonstrated FedALS's effectiveness in heterogeneous setups.

### Acknowledgments

This work was supported in parts by the Army Research Lab (W911NF2420172), and the National Science Foundation (CCF-1942878, CNS-2148182, CNS-2112471).

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

## A  Appendix

## B  Proof of Theorem 4.1

We first state and prove the following lemma that will be used in the proof of Theorem 4.1.

**Lemma B.1** (Leave-one-out). *[Expansion of Theorem 1 in Barnes et al. (2022)]*

*Let $\boldsymbol{S}'_k = (\boldsymbol{z}'_{k,1}, ..., \boldsymbol{z}'_{k,n_k})$, where $\boldsymbol{z}'_{k,i}$ is sampled from $\mathcal{D}_k$. Denote $\boldsymbol{S}^{(k)} = (\boldsymbol{S}_1, ..., \boldsymbol{S}'_k, ..., \boldsymbol{S}_K)$. Then*

$$\mathbb{E}_{\{\boldsymbol{S}_k \sim \mathcal{D}_k^{n_k}\}_{k=1}^K} \Delta_{\mathcal{A}}(\boldsymbol{S}) = \mathbb{E}_{k \sim \mathcal{K}, \{\boldsymbol{S}_k, \boldsymbol{S}'_k \sim \mathcal{D}_k^{n_k}\}_{k=1}^K} \left[ \frac{1}{n_k} \sum_{i=1}^{n_k} \left( l(\mathcal{A}(\boldsymbol{S}), \boldsymbol{z}'_{k,i}) - l(\mathcal{A}(\boldsymbol{S}^{(k)}), \boldsymbol{z}'_{k,i}) \right) \right]. \tag{10}$$

*Proof.* We have

$$\mathbb{E}_{\{\boldsymbol{S}_k \sim \mathcal{D}_k^{n_k}\}_{k=1}^K} R(\mathcal{A}(\boldsymbol{S})) = \mathbb{E}_{k \sim \mathcal{K}, \{\boldsymbol{S}_k, \boldsymbol{S}'_k \sim \mathcal{D}_k^{n_k}\}_{k=1}^K} l(\mathcal{A}(\boldsymbol{S}), \boldsymbol{z}'_{k,i}). \tag{11}$$

Also, observe that

$$\mathbb{E}_{\{\boldsymbol{S}_k \sim \mathcal{D}_k^{n_k}\}_{k=1}^K} R_{\boldsymbol{S}}(\mathcal{A}(\boldsymbol{S})) = \mathbb{E}_{k \sim \mathcal{K}, \{\boldsymbol{S}_k \sim \mathcal{D}_k^{n_k}\}_{k=1}^K} \left[ \frac{1}{n_k} \sum_{i=1}^{n_k} l(\mathcal{A}(\boldsymbol{S}), \boldsymbol{z}_{k,i}) \right] \tag{12}$$

$$= \mathbb{E}_{k \sim \mathcal{K}, \{\boldsymbol{S}_k, \boldsymbol{S}'_k \sim \mathcal{D}_k^{n_k}\}_{k=1}^K} \left[ \frac{1}{n_k} \sum_{i=1}^{n_k} l(\mathcal{A}(\boldsymbol{S}^{(k)}), \boldsymbol{z}'_{k,i}) \right]. \tag{13}$$

Putting 11, and 13 together, and by the definition of the expected generalization error, we get

$$\mathbb{E}_{\{\boldsymbol{S}_k \sim \mathcal{D}_k^{n_k}\}_{k=1}^K} \Delta_{\mathcal{A}}(\boldsymbol{S}) = \mathbb{E}_{k \sim \mathcal{K}, \{\boldsymbol{S}_k, \boldsymbol{S}'_k \sim \mathcal{D}_k^{n_k}\}_{k=1}^K} \left[ \frac{1}{n_k} \sum_{i=1}^{n_k} \left( l(\mathcal{A}(\boldsymbol{S}), \boldsymbol{z}'_{k,i}) - l(\mathcal{A}(\boldsymbol{S}^{(k)}), \boldsymbol{z}'_{k,i}) \right) \right]. \tag{14}$$

$\square$

In the following lemma, we establish a fundamental generalization bound for a single round of ERM and FedAvg. (Theorem 4.1).

**Theorem B.2.** *Let $l(M_{\boldsymbol{\theta}}, \boldsymbol{z})$ be $\mu$-strongly convex and $L$-smooth in $M_{\boldsymbol{\theta}}$, $M_{\boldsymbol{\theta}_k} = \mathcal{A}_k(\boldsymbol{S}_k)$ represents the model obtained from Empirical Risk Minimization (ERM) algorithm on local dataset $\boldsymbol{S}_k$, i.e., $M_{\boldsymbol{\theta}_k} = \arg\min_M \sum_{i=1}^{n_k} l(M, \boldsymbol{z}_{k,i})$, and $M_{\hat{\boldsymbol{\theta}}} = \mathcal{A}(\boldsymbol{S})$ is the model after one round of FedAvg ($\hat{\boldsymbol{\theta}} = \mathbb{E}_{k \sim \mathcal{K}} \boldsymbol{\theta}_k$). Then, the expected generalization error, $\mathbb{E}_{\{\boldsymbol{S}_k \sim \mathcal{D}_k^{n_k}\}_{k=1}^K} \Delta_{\mathcal{A}}(\boldsymbol{S})$, is bounded by*

$$\mathbb{E}_{k \sim \mathcal{K}} \left[ \frac{L\mathcal{K}(k)^2}{\mu} \underbrace{\mathbb{E}_{\{\boldsymbol{S}_k \sim \mathcal{D}_k^{n_k}\}} \Delta_{\mathcal{A}_k}(\boldsymbol{S}_k)}_{Expected\ local\ generalization} + 2\sqrt{\frac{L}{\mu}} \mathcal{K}(k) \underbrace{\sqrt{\mathbb{E}_{\{\boldsymbol{S}_k \sim \mathcal{D}_k^{n_k}\}_{k=1}^K} \delta_{k,\mathcal{A}}(\boldsymbol{S})}}_{Root\ of\ expected\ non\text{-}iidness} \underbrace{\sqrt{\mathbb{E}_{\{\boldsymbol{S}_k \sim \mathcal{D}_k^{n_k}\}} \Delta_{\mathcal{A}_k}(\boldsymbol{S}_k)}}_{Root\ of\ expected\ local\ generalization} \right],$$

$$\tag{15}$$

*where $\delta_{k,\mathcal{A}}(\boldsymbol{S}) = \left[ R_{\boldsymbol{S}_k}(\mathcal{A}(\boldsymbol{S})) - R_{\boldsymbol{S}_k}(\mathcal{A}_k(\boldsymbol{S}_k)) \right]$ indicates the level of non-iidness for client $k$ in function $\mathcal{A}$ on dataset $\boldsymbol{S}$.*

*Proof.* We again consider $\boldsymbol{S}'_k = (\boldsymbol{z}'_{k,1}, ..., \boldsymbol{z}'_{k,n_k})$, where $\boldsymbol{z}'_{k,i}$ is sampled from $\mathcal{D}_k$. Let also define $\boldsymbol{S}^{(k)} = (\boldsymbol{S}_1, ..., \boldsymbol{S}'_k, ..., \boldsymbol{S}_K)$. Based on Lemma B.1, we can express the expected generalization error as

$$\mathbb{E}_{\{\boldsymbol{S}_k \sim \mathcal{D}_k^{n_k}\}_{k=1}^K} \Delta_{\mathcal{A}}(\boldsymbol{S}) = \mathbb{E}_{k \sim \mathcal{K}, \{\boldsymbol{S}_k, \boldsymbol{S}'_k \sim \mathcal{D}_k^{n_k}\}_{k=1}^K} \left[ \frac{1}{n_k} \sum_{i=1}^{n_k} \left( l(\mathcal{A}(\boldsymbol{S}), \boldsymbol{z}'_{k,i}) - l(\mathcal{A}(\boldsymbol{S}^{(k)}), \boldsymbol{z}'_{k,i}) \right) \right]. \tag{16}$$

Based on $L$-smoothness of $l(M_{\boldsymbol{\theta}}, \boldsymbol{z})$ in $M_{\boldsymbol{\theta}}$, we obtain

$$\frac{1}{n_k}\sum_{i=1}^{n_k}\left(l(\mathcal{A}(\boldsymbol{S}), \boldsymbol{z}'_{k,i}) - l(\mathcal{A}(\boldsymbol{S}^{(k)}), \boldsymbol{z}'_{k,i})\right) \le \langle \nabla \frac{1}{n_k}\sum_{i=1}^{n_k} l(\mathcal{A}(\boldsymbol{S}^{(k)}), \boldsymbol{z}'_{k,i}), \mathcal{A}(\boldsymbol{S}) - \mathcal{A}(\boldsymbol{S}^{(k)})\rangle + \frac{L}{2}\|\mathcal{A}(\boldsymbol{S}) - \mathcal{A}(\boldsymbol{S}^{(k)})\|^2,$$
(17)

where $\langle\cdot,\cdot\rangle, \|\cdot\|^2$ indicate Euclidean inner product, and squared $L$2-norm. Note that (17) holds due to

$$f(\boldsymbol{y}) \le f(\boldsymbol{x}) + \langle \nabla f(\boldsymbol{x}), \boldsymbol{y} - \boldsymbol{x}\rangle + \frac{L}{2}\|\boldsymbol{y} - \boldsymbol{x}\|^2.$$
(18)

We can bound expectation of the inner product term on the right-hand side of (17) using Cauchy–Schwarz inequality as

$$\mathbb{E}_{k\sim\mathcal{K}, \{\boldsymbol{S}_k, \boldsymbol{S}'_k \sim \mathcal{D}_k^{n_k}\}_{k=1}^K} \langle \nabla \frac{1}{n_k}\sum_{i=1}^{n_k} l(\mathcal{A}(\boldsymbol{S}^{(k)}), \boldsymbol{z}'_{k,i}), \mathcal{A}(\boldsymbol{S}) - \mathcal{A}(\boldsymbol{S}^{(k)})\rangle$$

$$\le \mathbb{E}_{k\sim\mathcal{K}} \mathbb{E}_{\{\boldsymbol{S}_k, \boldsymbol{S}'_k \sim \mathcal{D}_k^{n_k}\}_{k=1}^K} |\langle \nabla \frac{1}{n_k}\sum_{i=1}^{n_k} l(\mathcal{A}(\boldsymbol{S}^{(k)}), \boldsymbol{z}'_{k,i}), \mathcal{A}(\boldsymbol{S}) - \mathcal{A}(\boldsymbol{S}^{(k)})\rangle|$$
(19)

$$\le \mathbb{E}_{k\sim\mathcal{K}}\left[\mathbb{E}_{\{\boldsymbol{S}_k, \boldsymbol{S}'_k \sim \mathcal{D}_k^{n_k}\}_{k=1}^K} \|\nabla \frac{1}{n_k}\sum_{i=1}^{n_k} l(\mathcal{A}(\boldsymbol{S}^{(k)}), \boldsymbol{z}'_{k,i})\| \quad \|\mathcal{A}(\boldsymbol{S}) - \mathcal{A}(\boldsymbol{S}^{(k)})\|\right]$$
(20)

$$\le \mathbb{E}_{k\sim\mathcal{K}}\sqrt{\mathbb{E}_{\{\boldsymbol{S}_k, \boldsymbol{S}'_k \sim \mathcal{D}_k^{n_k}\}_{k=1}^K} \|\nabla \frac{1}{n_k}\sum_{i=1}^{n_k} l(\mathcal{A}(\boldsymbol{S}^{(k)}), \boldsymbol{z}'_{k,i})\|^2 \, \mathbb{E}_{\{\boldsymbol{S}_k, \boldsymbol{S}'_k \sim \mathcal{D}_k^{n_k}\}_{k=1}^K} \|\mathcal{A}(\boldsymbol{S}) - \mathcal{A}(\boldsymbol{S}^{(k)})\|^2}, \quad (21)$$

where (19) is true because on the right we have the absolute value. (20), and (21) are based on Cauchy–Schwarz inequality.

Now Let's find an upper bound for $\mathbb{E}_{\{\boldsymbol{S}_k, \boldsymbol{S}'_k \sim \mathcal{D}_k^{n_k}\}_{k=1}^K} \|\mathcal{A}(\boldsymbol{S}) - \mathcal{A}(\boldsymbol{S}^{(k)})\|^2$ that appears on the right-hand side of both (17), and (21). We obtain

$$\mathbb{E}_{\{\boldsymbol{S}_k, \boldsymbol{S}'_k \sim \mathcal{D}_k^{n_k}\}_{k=1}^K} \|\mathcal{A}(\boldsymbol{S}) - \mathcal{A}(\boldsymbol{S}^{(k)})\|^2$$

$$= \mathbb{E}_{\{\boldsymbol{S}_k, \boldsymbol{S}'_k \sim \mathcal{D}_k^{n_k}\}_{k=1}^K} \mathcal{K}(k)^2 \|\mathcal{A}_k(\boldsymbol{S}_k) - \mathcal{A}_k(\boldsymbol{S}'_k)\|^2 \quad (22)$$

$$\le \mathbb{E}_{\{\boldsymbol{S}_k, \boldsymbol{S}'_k \sim \mathcal{D}_k^{n_k}\}_{k=1}^K} \frac{2\mathcal{K}(k)^2}{\mu}\left(R_{\boldsymbol{S}'_k}(\mathcal{A}_k(\boldsymbol{S}_k)) - R_{\boldsymbol{S}'_k}(\mathcal{A}_k(\boldsymbol{S}'_k))\right) \quad (23)$$

$$= \mathbb{E}_{\{\boldsymbol{S}_k, \boldsymbol{S}'_k \sim \mathcal{D}_k^{n_k}\}_{k=1}^K} \frac{2\mathcal{K}(k)^2}{\mu} \frac{1}{n_k}\sum_{j=1}^{n_k}\left(l(\mathcal{A}_k(\boldsymbol{S}_k), \boldsymbol{z}'_{k,j}) - l(\mathcal{A}_k(\boldsymbol{S}'_k), \boldsymbol{z}'_{k,j})\right) \quad (24)$$

$$= \mathbb{E}_{\{\boldsymbol{S}_k, \boldsymbol{S}'_k \sim \mathcal{D}_k^{n_k}\}_{k=1}^K} \frac{2\mathcal{K}(k)^2}{\mu}\Delta_{\mathcal{A}_k}(\boldsymbol{S}'_k) \quad (25)$$

$$= \mathbb{E}_{\{\boldsymbol{S}_k \sim \mathcal{D}_k^{n_k}\}_{k=1}^K} \frac{2\mathcal{K}(k)^2}{\mu}\Delta_{\mathcal{A}_k}(\boldsymbol{S}_k), \quad (26)$$

where (22) proceeds by observing that $\mathcal{A}(\boldsymbol{S}^{(k,i)})$ varies solely in the sub-model derived from node $k$, diverging from $\mathcal{A}(\boldsymbol{S})$, and this discrepancy is magnified by a factor of $\mathcal{K}(k)$ when averaging of all sub-models. (23) holds due to the $\mu$-strongly convexity of $l(M_{\boldsymbol{\theta}}, \boldsymbol{z})$ in $M_{\boldsymbol{\theta}}$ which leads to $\mu$-strongly convexity of $R_{\boldsymbol{S}_k}(M_{\boldsymbol{\theta}})$ and the fact that $\mathcal{A}_k(\boldsymbol{S}'_k)$ is derived from the local ERM, *i.e.*, $\mathcal{A}_k(\boldsymbol{S}'_k) = \arg\min_M\left(\sum_{i=1}^{n_k} l(M, \boldsymbol{z}'_{k,i})\right)$ and $\nabla R_{\boldsymbol{S}'_k}(\mathcal{A}_k(\boldsymbol{S}'_k)) = 0$. Note that if $f$ is $\mu$-strongly convex, we get

$$f(\boldsymbol{x}) - f(\boldsymbol{y}) + \frac{\mu}{2}\|\boldsymbol{x} - \boldsymbol{y}\|^2 \le \langle \nabla f(\boldsymbol{x}), \boldsymbol{x} - \boldsymbol{y}\rangle. \quad (27)$$

(24), (25) are based on local empirical and population risk definitions.

Now we bound $\mathbb{E}_{k\sim\mathcal{K},\{\boldsymbol{S}_k,\boldsymbol{S}'_k\sim\mathcal{D}_k^{n_k}\}_{k=1}^K}\|\nabla\frac{1}{n_k}\sum_{i=1}^{n_k}l(\mathcal{A}(\boldsymbol{S}^{(k)}),\boldsymbol{z}'_{k,i})\|^2$ on the right-hand side of (21). Note that

$$\|\nabla\frac{1}{n_k}\sum_{i=1}^{n_k}l(\mathcal{A}(\boldsymbol{S}^{(k)}),\boldsymbol{z}'_{k,i})\|^2 \leq 2L\left(\frac{1}{n_k}\sum_{i=1}^{n_k}l(\mathcal{A}(\boldsymbol{S}^{(k)}),\boldsymbol{z}'_{k,i})-\frac{1}{n_k}\sum_{i=1}^{n_k}l(\mathcal{A}_k(\boldsymbol{S}'_k),\boldsymbol{z}'_{k,i})\right) \tag{28}$$

$$\leq 2L\left(R_{\boldsymbol{S}'_k}(\mathcal{A}(\boldsymbol{S}^{(k)}))-R_{\boldsymbol{S}'_k}(\mathcal{A}_k(\boldsymbol{S}'_k))\right), \tag{29}$$

where 28 is obtained using the fact that for any $L$-smooth function $f$, we have

$$\|\nabla f(\boldsymbol{x})\|^2 \leq 2L(f(\boldsymbol{x})-f^*), \tag{30}$$

and the fact that $\mathcal{A}_k(\boldsymbol{S}'_k)$ is derived from the local ERM, *i.e.*, $\mathcal{A}_k(\boldsymbol{S}'_k)=\arg\min_M\sum_{i=1}^{n_k}l(M,\boldsymbol{z}'_{k,i})$. 29 is based on the definition of local empirical risk.

Putting (17) into (16) and considering (21) we get

$$\mathbb{E}_{\{\boldsymbol{S}_k\sim\mathcal{D}_k^{n_k}\}_{k=1}^K}\Delta_{\mathcal{A}}(\boldsymbol{S})$$

$$\leq \mathbb{E}_{k\sim\mathcal{K}}\left[\mathbb{E}_{\{\boldsymbol{S}_k,\boldsymbol{S}'_k\sim\mathcal{D}_k^{n_k}\}_{k=1}^K}\frac{L}{2}\|\mathcal{A}(\boldsymbol{S})-\mathcal{A}(\boldsymbol{S}^{(k)})\|^2\right. \tag{31}$$

$$\left.+\sqrt{\mathbb{E}_{\{\boldsymbol{S}_k,\boldsymbol{S}'_k\sim\mathcal{D}_k^{n_k}\}_{k=1}^K}\|\nabla\frac{1}{n_k}\sum_{i=1}^{n_k}l(\mathcal{A}(\boldsymbol{S}^{(k)}),\boldsymbol{z}'_{k,i})\|^2\,\mathbb{E}_{\{\boldsymbol{S}_k,\boldsymbol{S}'_k\sim\mathcal{D}_k^{n_k}\}_{k=1}^K}\|\mathcal{A}(\boldsymbol{S})-\mathcal{A}(\boldsymbol{S}^{(k)})\|^2}\right]$$

$$\leq \mathbb{E}_{k\sim\mathcal{K}}\left[\mathbb{E}_{\{\boldsymbol{S}_k\sim\mathcal{D}_k^{n_k}\}_{k=1}^K}\frac{L\mathcal{K}(k)^2}{\mu}\Delta_{\mathcal{A}_k}(\boldsymbol{S}_k)\right. \tag{32}$$

$$\left.+\sqrt{\mathbb{E}_{\{\boldsymbol{S}_k,\boldsymbol{S}'_k\sim\mathcal{D}_k^{n_k}\}_{k=1}^K}2L\left(R_{\boldsymbol{S}'_k}(\mathcal{A}(\boldsymbol{S}^{(k)}))-R_{\boldsymbol{S}'_k}(\mathcal{A}_k(\boldsymbol{S}'_k))\right)\mathbb{E}_{\{\boldsymbol{S}_k\sim\mathcal{D}_k^{n_k}\}_{k=1}^K}\frac{2\mathcal{K}(k)^2}{\mu}\Delta_{\mathcal{A}_k}(\boldsymbol{S}_k)}\right]$$

$$\leq \mathbb{E}_{k\sim\mathcal{K}}\left[\mathbb{E}_{\{\boldsymbol{S}_k\sim\mathcal{D}_k^{n_k}\}_{k=1}^K}\frac{L\mathcal{K}(k)^2}{\mu}\Delta_{\mathcal{A}_k}(\boldsymbol{S}_k)\right. \tag{33}$$

$$\left.+\sqrt{\mathbb{E}_{\{\boldsymbol{S}_k\sim\mathcal{D}_k^{n_k}\}_{k=1}^K}2L\left(R_{\boldsymbol{S}_k}(\mathcal{A}(\boldsymbol{S}))-R_{\boldsymbol{S}_k}(\mathcal{A}_k(\boldsymbol{S}_k))\right)\mathbb{E}_{\{\boldsymbol{S}_k\sim\mathcal{D}_k^{n_k}\}_{k=1}^K}\frac{2\mathcal{K}(k)^2}{\mu}\Delta_{\mathcal{A}_k}(\boldsymbol{S}_k)}\right]$$

$$\leq \mathbb{E}_{k\sim\mathcal{K}}\left[\frac{L\mathcal{K}(k)^2}{\mu}\mathbb{E}_{\{\boldsymbol{S}_k\sim\mathcal{D}_k^{n_k}\}_{k=1}^K}\Delta_{\mathcal{A}_k}(\boldsymbol{S}_k)+2\sqrt{\frac{L}{\mu}}\mathcal{K}(k)\sqrt{\mathbb{E}_{\{\boldsymbol{S}_k\sim\mathcal{D}_k^{n_k}\}_{k=1}^K}\delta_{k,\mathcal{A}}(\boldsymbol{S})\,\mathbb{E}_{\{\boldsymbol{S}_k\sim\mathcal{D}_k^{n_k}\}_{k=1}^K}\Delta_{\mathcal{A}_k}(\boldsymbol{S}_k)}\right]$$

$$\tag{34}$$

$$\tag{35}$$

,

where in (32) we have applied (26), and (29). (33) proceeds by considering that

$$\mathbb{E}_{\{\boldsymbol{S}_k,\boldsymbol{S}'_k\sim\mathcal{D}_k^{n_k}\}_{k=1}^K}\left[R_{\boldsymbol{S}'_k}(\mathcal{A}(\boldsymbol{S}^{(k)}))-R_{\boldsymbol{S}'_k}(\mathcal{A}_k(\boldsymbol{S}'_k))\right]=\mathbb{E}_{\{\boldsymbol{S}_k\sim\mathcal{D}_k^{n_k}\}_{k=1}^K}\left[R_{\boldsymbol{S}_k}(\mathcal{A}(\boldsymbol{S}))-R_{\boldsymbol{S}_k}(\mathcal{A}_k(\boldsymbol{S}_k))\right]. \tag{36}$$

In (34) we have used the definition of $\delta_{k,\mathcal{A}}(\boldsymbol{S})=\left[R_{\boldsymbol{S}_k}(\mathcal{A}(\boldsymbol{S}))-R_{\boldsymbol{S}_k}(\mathcal{A}_k(\boldsymbol{S}_k))\right]$. This completes the proof and provides the following upper bound for $\mathbb{E}_{\{\boldsymbol{S}_k\sim\mathcal{D}_k^{n_k}\}_{k=1}^K}\Delta_{\mathcal{A}}(\boldsymbol{S})$,

$$\mathbb{E}_{k\sim\mathcal{K}}\left[\frac{L\mathcal{K}(k)^2}{\mu}\mathbb{E}_{\{\boldsymbol{S}_k\sim\mathcal{D}_k^{n_k}\}}\Delta_{\mathcal{A}_k}(\boldsymbol{S}_k)+2\sqrt{\frac{L}{\mu}}\mathcal{K}(k)\sqrt{\mathbb{E}_{\{\boldsymbol{S}_k\sim\mathcal{D}_k^{n_k}\}_{k=1}^K}\delta_{k,\mathcal{A}}(\boldsymbol{S})\,\mathbb{E}_{\{\boldsymbol{S}_k\sim\mathcal{D}_k^{n_k}\}}\Delta_{\mathcal{A}_k}(\boldsymbol{S}_k)}\right]. \tag{37}$$

$\square$

## C   Proof of Theorem 4.3

Here we provide an identical theorem as Theorem B.2, except that instead of ERM, multiple local stochastic gradient descent steps are used as the local optimizer.

**Theorem C.1.** *Let $l(M_{\boldsymbol{\theta}}, \boldsymbol{z})$ be $\mu$-strongly convex and $L$-smooth in $M_{\boldsymbol{\theta}}$, $M_{\boldsymbol{\theta}_k} = \mathcal{A}_k(\boldsymbol{S}_k)$ represents the model obtained by doing multiple local steps as in (5) on local dataset $\boldsymbol{S}_k$, and $M_{\hat{\boldsymbol{\theta}}} = \mathcal{A}(\boldsymbol{S})$ is the model after one round of FedAvg ($\hat{\boldsymbol{\theta}} = \mathbb{E}_{k \sim \mathcal{K}} \boldsymbol{\theta}_k$). Then, the expected generalization error, $\mathbb{E}_{\{\boldsymbol{S}_k \sim \mathcal{D}_k^{n_k}\}_{k=1}^K} \Delta_{\mathcal{A}}(\boldsymbol{S})$, is bounded by*

$$\mathbb{E}_{k \sim \mathcal{K}} \left[ \mathbb{E}_{\{\boldsymbol{S}_k \sim \mathcal{D}_k^{n_k}\}_{k=1}^K} \frac{2L\mathcal{K}(k)^2}{\mu} \left( \Delta_{\mathcal{A}_k}(\boldsymbol{S}_k) + 2\epsilon_k(\boldsymbol{S}_k) \right) \right. \tag{38}$$
$$\left. + \sqrt{\frac{8L}{\mu}} \mathcal{K}(k) \sqrt{\mathbb{E}_{\{\boldsymbol{S}_k \sim \mathcal{D}_k^{n_k}\}_{k=1}^K} \left( \delta_{k,\mathcal{A}}(\boldsymbol{S}) + \epsilon_k(\boldsymbol{S}_k) \right) \mathbb{E}_{\{\boldsymbol{S}_k \sim \mathcal{D}_k^{n_k}\}_{k=1}^K} \left( \Delta_{\mathcal{A}_k}(\boldsymbol{S}_k) + 2\epsilon_k(\boldsymbol{S}_k) \right)} \right],$$

*where $\epsilon_k(\boldsymbol{S}_k) = R_{\boldsymbol{S}_k}(\mathcal{A}_k(\boldsymbol{S}_k)) - R_{\boldsymbol{S}_k}(\mathcal{A}^*(\boldsymbol{S}_k))$.*

*Proof.* All the steps are exactly the same as in the proof of theorem B.2 except for the two steps below:

First, we use a new upper bound for $\mathbb{E}_{\{\boldsymbol{S}_k, \boldsymbol{S}_k' \sim \mathcal{D}_k^{n_k}\}_{k=1}^K} \|\mathcal{A}(\boldsymbol{S}) - \mathcal{A}(\boldsymbol{S}^{(k)})\|^2$ that appears on the right-hand side of both (17), and (21). We have

$$\mathbb{E}_{\{\boldsymbol{S}_k, \boldsymbol{S}_k' \sim \mathcal{D}_k^{n_k}\}_{k=1}^K} \|\mathcal{A}(\boldsymbol{S}) - \mathcal{A}(\boldsymbol{S}^{(k)})\|^2$$

$$= \mathbb{E}_{\{\boldsymbol{S}_k, \boldsymbol{S}_k' \sim \mathcal{D}_k^{n_k}\}_{k=1}^K} \mathcal{K}(k)^2 \|\mathcal{A}_k(\boldsymbol{S}_k) - \mathcal{A}_k(\boldsymbol{S}_k')\|^2 \tag{39}$$

$$= \mathbb{E}_{\{\boldsymbol{S}_k, \boldsymbol{S}_k' \sim \mathcal{D}_k^{n_k}\}_{k=1}^K} \mathcal{K}(k)^2 \|\mathcal{A}_k(\boldsymbol{S}_k) - \mathcal{A}^*(\boldsymbol{S}_k') + \mathcal{A}^*(\boldsymbol{S}_k') - \mathcal{A}_k(\boldsymbol{S}_k')\|^2 \tag{40}$$

$$= \mathbb{E}_{\{\boldsymbol{S}_k, \boldsymbol{S}_k' \sim \mathcal{D}_k^{n_k}\}_{k=1}^K} 2\mathcal{K}(k)^2 \left( \|\mathcal{A}_k(\boldsymbol{S}_k) - \mathcal{A}^*(\boldsymbol{S}_k')\|^2 + \|\mathcal{A}^*(\boldsymbol{S}_k') - \mathcal{A}_k(\boldsymbol{S}_k')\|^2 \right) \tag{41}$$

$$\leq \mathbb{E}_{\{\boldsymbol{S}_k, \boldsymbol{S}_k' \sim \mathcal{D}_k^{n_k}\}_{k=1}^K} \frac{4\mathcal{K}(k)^2}{\mu} \left( R_{\boldsymbol{S}_k'}(\mathcal{A}_k(\boldsymbol{S}_k)) - R_{\boldsymbol{S}_k'}(\mathcal{A}^*(\boldsymbol{S}_k')) + R_{\boldsymbol{S}_k'}(\mathcal{A}_k(\boldsymbol{S}_k')) - R_{\boldsymbol{S}_k'}(\mathcal{A}^*(\boldsymbol{S}_k')) \right) \tag{42}$$

$$= \mathbb{E}_{\{\boldsymbol{S}_k, \boldsymbol{S}_k' \sim \mathcal{D}_k^{n_k}\}_{k=1}^K} \frac{4\mathcal{K}(k)^2}{\mu} \left( R_{\boldsymbol{S}_k'}(\mathcal{A}_k(\boldsymbol{S}_k)) - R_{\boldsymbol{S}_k'}(\mathcal{A}_k(\boldsymbol{S}_k')) + 2R_{\boldsymbol{S}_k'}(\mathcal{A}_k(\boldsymbol{S}_k')) - 2R_{\boldsymbol{S}_k'}(\mathcal{A}^*(\boldsymbol{S}_k')) \right) \tag{43}$$

$$= \mathbb{E}_{\{\boldsymbol{S}_k, \boldsymbol{S}_k' \sim \mathcal{D}_k^{n_k}\}_{k=1}^K} \frac{4\mathcal{K}(k)^2}{\mu} \left( \frac{1}{n_k} \sum_{j=1}^{n_k} \left( l(\mathcal{A}_k(\boldsymbol{S}_k), \boldsymbol{z}_{k,j}') - l(\mathcal{A}_k(\boldsymbol{S}_k'), \boldsymbol{z}_{k,j}') \right) + 2R_{\boldsymbol{S}_k'}(\mathcal{A}_k(\boldsymbol{S}_k')) - 2R_{\boldsymbol{S}_k'}(\mathcal{A}^*(\boldsymbol{S}_k')) \right)$$
$$\tag{44}$$

$$= \mathbb{E}_{\{\boldsymbol{S}_k, \boldsymbol{S}_k' \sim \mathcal{D}_k^{n_k}\}_{k=1}^K} \frac{4\mathcal{K}(k)^2}{\mu} \left( \Delta_{\mathcal{A}_k}(\boldsymbol{S}_k') + 2R_{\boldsymbol{S}_k'}(\mathcal{A}_k(\boldsymbol{S}_k')) - 2R_{\boldsymbol{S}_k'}(\mathcal{A}^*(\boldsymbol{S}_k')) \right) \tag{45}$$

$$= \mathbb{E}_{\{\boldsymbol{S}_k \sim \mathcal{D}_k^{n_k}\}_{k=1}^K} \frac{4\mathcal{K}(k)^2}{\mu} \left( \Delta_{\mathcal{A}_k}(\boldsymbol{S}_k) + 2R_{\boldsymbol{S}_k}(\mathcal{A}_k(\boldsymbol{S}_k)) - 2R_{\boldsymbol{S}_k}(\mathcal{A}^*(\boldsymbol{S}_k)) \right), \tag{46}$$

where in (40) $\mathcal{A}^*(\boldsymbol{S}_k)$ is the ERM on $\boldsymbol{S}_k$. (41) is based on the following inequality.

$$\|\sum_{i=1}^n a_i\|^2 \leq n \sum_{i=1}^n \|a_i\|^2. \tag{47}$$

Secondly, we derive a new bound for $\mathbb{E}_{k\sim\mathcal{K},\{\boldsymbol{S}_k,\boldsymbol{S}'_k\sim\mathcal{D}_k^{n_k}\}_{k=1}^K}\|\nabla\frac{1}{n_k}\sum_{i=1}^{n_k}l(\mathcal{A}(\boldsymbol{S}^{(k)}),\boldsymbol{z}'_{k,i})\|^2$ on the right hand side of (21). We get

$$\|\nabla\frac{1}{n_k}\sum_{i=1}^{n_k}l(\mathcal{A}(\boldsymbol{S}^{(k)}),\boldsymbol{z}'_{k,i})\|^2 \leq 2L\left(\frac{1}{n_k}\sum_{i=1}^{n_k}l(\mathcal{A}(\boldsymbol{S}^{(k)}),\boldsymbol{z}'_{k,i})-\frac{1}{n_k}\sum_{i=1}^{n_k}l(\mathcal{A}^*(\boldsymbol{S}'_k),\boldsymbol{z}'_{k,i})\right) \tag{48}$$

$$\leq 2L\left(R_{\boldsymbol{S}'_k}(\mathcal{A}(\boldsymbol{S}^{(k)}))-R_{\boldsymbol{S}'_k}(\mathcal{A}^*(\boldsymbol{S}'_k))\right) \tag{49}$$

$$\leq 2L\left(R_{\boldsymbol{S}'_k}(\mathcal{A}(\boldsymbol{S}^{(k)}))-R_{\boldsymbol{S}'_k}(\mathcal{A}_k(\boldsymbol{S}'_k))+R_{\boldsymbol{S}'_k}(\mathcal{A}_k(\boldsymbol{S}'_k))-R_{\boldsymbol{S}'_k}(\mathcal{A}^*(\boldsymbol{S}'_k))\right) \tag{50}$$

$$\leq 2L\left(\delta_{k,\mathcal{A}}(\boldsymbol{S}^{(k)})+R_{\boldsymbol{S}'_k}(\mathcal{A}_k(\boldsymbol{S}'_k))-R_{\boldsymbol{S}'_k}(\mathcal{A}^*(\boldsymbol{S}'_k))\right). \tag{51}$$

Let's define $\epsilon_k(\boldsymbol{S}_k)=R_{\boldsymbol{S}_k}(\mathcal{A}_k(\boldsymbol{S}_k))-R_{\boldsymbol{S}_k}(\mathcal{A}^*(\boldsymbol{S}_k))$. Putting (17) into (16) and considering (21) we get

$$\mathbb{E}_{\{\boldsymbol{S}_k\sim\mathcal{D}_k^{n_k}\}_{k=1}^K}\Delta_\mathcal{A}(\boldsymbol{S})\leq\mathbb{E}_{k\sim\mathcal{K}}\Bigg[\mathbb{E}_{\{\boldsymbol{S}_k,\boldsymbol{S}'_k\sim\mathcal{D}_k^{n_k}\}_{k=1}^K}\frac{L}{2}\|\mathcal{A}(\boldsymbol{S})-\mathcal{A}(\boldsymbol{S}^{(k)})\|^2$$

$$+\sqrt{\mathbb{E}_{\{\boldsymbol{S}_k,\boldsymbol{S}'_k\sim\mathcal{D}_k^{n_k}\}_{k=1}^K}\|\nabla\frac{1}{n_k}\sum_{i=1}^{n_k}l(\mathcal{A}(\boldsymbol{S}^{(k)}),\boldsymbol{z}'_{k,i})\|^2\,\mathbb{E}_{\{\boldsymbol{S}_k,\boldsymbol{S}'_k\sim\mathcal{D}_k^{n_k}\}_{k=1}^K}\|\mathcal{A}(\boldsymbol{S})-\mathcal{A}(\boldsymbol{S}^{(k)})\|^2}\Bigg]$$

$$\leq\mathbb{E}_{k\sim\mathcal{K}}\Bigg[\mathbb{E}_{\{\boldsymbol{S}_k\sim\mathcal{D}_k^{n_k}\}_{k=1}^K}\frac{2L\mathcal{K}(k)^2}{\mu}\left(\Delta_{\mathcal{A}_k}(\boldsymbol{S}_k)+2\epsilon_k(\boldsymbol{S}_k)\right) \tag{52}$$

$$+\sqrt{\mathbb{E}_{\{\boldsymbol{S}_k,\boldsymbol{S}'_k\sim\mathcal{D}_k^{n_k}\}_{k=1}^K}2L\left(\delta_{k,\mathcal{A}}(\boldsymbol{S}^{(k)})+\epsilon_k(\boldsymbol{S}'_k)\right)\mathbb{E}_{\{\boldsymbol{S}_k\sim\mathcal{D}_k^{n_k}\}_{k=1}^K}\frac{4\mathcal{K}(k)^2}{\mu}\left(\Delta_{\mathcal{A}_k}(\boldsymbol{S}_k)+2\epsilon_k(\boldsymbol{S}_k)\right)}\Bigg]$$

$$\leq\mathbb{E}_{k\sim\mathcal{K}}\Bigg[\mathbb{E}_{\{\boldsymbol{S}_k\sim\mathcal{D}_k^{n_k}\}_{k=1}^K}\frac{2L\mathcal{K}(k)^2}{\mu}\left(\Delta_{\mathcal{A}_k}(\boldsymbol{S}_k)+2\epsilon_k(\boldsymbol{S}_k)\right) \tag{53}$$

$$+\sqrt{\frac{8L}{\mu}}\mathcal{K}(k)\sqrt{\mathbb{E}_{\{\boldsymbol{S}_k\sim\mathcal{D}_k^{n_k}\}_{k=1}^K}\left(\delta_{k,\mathcal{A}}(\boldsymbol{S})+\epsilon_k(\boldsymbol{S}_k)\right)\mathbb{E}_{\{\boldsymbol{S}_k\sim\mathcal{D}_k^{n_k}\}_{k=1}^K}\left(\Delta_{\mathcal{A}_k}(\boldsymbol{S}_k)+2\epsilon_k(\boldsymbol{S}_k)\right)}\Bigg],$$

where in (52) we have applied (46), and (51). This completes the proof. $\qquad\square$

Now, we prove Theorem 4.3 as follows.

**Theorem C.2.** *Let $l(M_{\boldsymbol{\theta}},\boldsymbol{z})$ be $\mu$-strongly convex and $L$-smooth in $M_{\boldsymbol{\theta}}$. Local models at round $r$ are calculated by doing $\tau$ local gradient descent steps (5), and the local gradient variance is bounded by $\sigma^2$, i.e., $\mathbb{E}_{\boldsymbol{z}\sim\mathcal{D}_k}\|\nabla l(M_{\boldsymbol{\theta}},\boldsymbol{z})-\mathbb{E}_{\boldsymbol{z}\sim\mathcal{D}_k}\nabla l(M_{\boldsymbol{\theta}},\boldsymbol{z})\|^2\leq\sigma^2$. The aggregated model at round $r$, $M_{\hat{\boldsymbol{\theta}}_r}$, is obtained by performing FedAvg, and the data points used in round $r$ (i.e., $Z_{k,r}$) are sampled without replacement. The average generalization error bound is*

$$\mathbb{E}_{\boldsymbol{S}}\,\Delta_{FedAvg}(\boldsymbol{S})\leq\frac{1}{R}\sum_{r=1}^R\mathbb{E}_{k\sim\mathcal{K}}\left[\frac{2L\mathcal{K}(k)^2}{\mu}A+\sqrt{\frac{8L}{\mu}}\mathcal{K}(k)(AB)^{\frac{1}{2}}\right]$$

*where $A=\tilde{O}\left(\sqrt{\frac{\mathcal{C}(M_{\boldsymbol{\theta}})}{|Z_{k,r}|}}+\frac{\sigma^2}{\mu\tau}\right)$, $B=\tilde{O}\left(\mathbb{E}_{\{Z_{k,r}\}_{k=1}^K}\delta_{k,\mathcal{A}}(\{Z_{k,r}\}_{k=1}^K)+\frac{\sigma^2}{\mu\tau}\right)$, $\tilde{O}$ hides constants and poly-logarithmic factors, and $\mathcal{C}(M_{\boldsymbol{\theta}})$ shows the complexity of the model class of $M_{\boldsymbol{\theta}}$.*

*Proof.* Based on the definition, we have

$$\mathbb{E}_{\{\boldsymbol{S}_k \sim \mathcal{D}_k^{n_k}\}_{k=1}^K} \Delta_{FedAvg}(\boldsymbol{S}) = \mathbb{E}_{\{\boldsymbol{S}_k \sim \mathcal{D}_k^{n_k}\}_{k=1}^K} \frac{1}{R} \sum_{r=1}^R \mathbb{E}_{k \sim \mathcal{K}} \left[ \mathbb{E}_{\boldsymbol{z} \sim \mathcal{D}_k} l(M_{\hat{\boldsymbol{\theta}}_r}, \boldsymbol{z}) - \frac{1}{|Z_{k,r}|} \sum_{i \in Z_{k,r}}^K l(M_{\hat{\boldsymbol{\theta}}_r}, \boldsymbol{z}_{k,i}) \right] \quad (54)$$

$$= \frac{1}{R} \sum_{r=1}^R \mathbb{E}_{\{Z_{k,r} \sim \mathcal{D}_k^{|Z_{k,r}|}\}_{k=1}^K} \mathbb{E}_{k \sim \mathcal{K}} \left[ \mathbb{E}_{\boldsymbol{z} \sim \mathcal{D}_k} l(M_{\hat{\boldsymbol{\theta}}_r}, \boldsymbol{z}) - \frac{1}{|Z_{k,r}|} \sum_{i \in Z_{k,r}}^K l(M_{\hat{\boldsymbol{\theta}}_r}, \boldsymbol{z}_{k,i}) \right] \quad (55)$$

$$= \frac{1}{R} \sum_{r=1}^R \mathbb{E}_{\{Z_{k,r} \sim \mathcal{D}_k^{|Z_{k,r}|}\}_{k=1}^K} \Delta_{\mathcal{A}}(\{Z_{k,r}\}_{k=1}^K) \quad (56)$$

$$\leq \frac{1}{R} \sum_{r=1}^R \mathbb{E}_{k \sim \mathcal{K}} \left[ \mathbb{E}_{\{Z_{k,r} \sim \mathcal{D}_k^{|Z_{k,r}|}\}} \frac{2LK(k)^2}{\mu} \left( \Delta_{\mathcal{A}_k}(Z_{k,r}) + 2\epsilon_k(Z_{k,r}) \right) \right. \quad (57)$$

$$\left. + \sqrt{\frac{8L}{\mu}} \mathcal{K}(k) \sqrt{\mathbb{E}_{\{Z_{k,r} \sim \mathcal{D}_k^{|Z_{k,r}|}\}_{k=1}^K} \left( \delta_{k,\mathcal{A}}(\{Z_{k,r}\}_{k=1}^K) + \epsilon_k(Z_{k,r}) \right) \mathbb{E}_{\{Z_{k,r} \sim \mathcal{D}_k^{|Z_{k,r}|}\}} \left( \Delta_{\mathcal{A}_k}(Z_{k,r}) + \epsilon_k(Z_{k,r}) \right)} \right]$$

$$\leq \frac{1}{R} \sum_{r=1}^R \mathbb{E}_{k \sim \mathcal{K}} \left[ \frac{2LK(k)^2}{\mu} \tilde{O}\left( \sqrt{\frac{\mathcal{C}(M_{\boldsymbol{\theta}})}{|Z_{k,r}|}} + \frac{\sigma^2}{\mu\tau} \right) \right. \quad (58)$$

$$\left. + \sqrt{\frac{8L}{\mu}} \mathcal{K}(k) \sqrt{\tilde{O}\left( \mathbb{E}_{\{Z_{k,r} \sim \mathcal{D}_k^{|Z_{k,r}|}\}_{k=1}^K} \delta_{k,\mathcal{A}}(\{Z_{k,r}\}_{k=1}^K) + \frac{\sigma^2}{\mu\tau} \right) \tilde{O}\left( \sqrt{\frac{\mathcal{C}(M_{\boldsymbol{\theta}})}{|Z_{k,r}|}} + \frac{\sigma^2}{\mu\tau} \right)} \right],$$

where in (56), $\mathcal{A}$ represents one-round FedAvg algorithm. In (57) we have used Theorem C.1. In (58) we have used the conventional statistical learning theory originated with Leslie Valiant's probably approximately correct (PAC) framework Valiant (1984). We have also applied the optimization convergence rate bounds in the literature Stich & Karimireddy (2019). Note that $\tilde{O}$ hides constants and poly-logarithmic factors. $\qquad\square$

## D  Partial Client Participation Setting

We first define an empirical risk for the partial participation distribution $\hat{\mathcal{K}}$ on dataset $\boldsymbol{S}$, where $Supp(\hat{\mathcal{K}}) \neq \{1, \ldots, K\}$ and $|Supp(\hat{\mathcal{K}})| = \hat{K} \leq K$, as

$$R_{\boldsymbol{S}}^{\hat{\mathcal{K}}}(M_{\boldsymbol{\theta}}) = \mathbb{E}_{k \sim \hat{\mathcal{K}}} R_{\boldsymbol{S}_k}(M_{\boldsymbol{\theta}}) = \mathbb{E}_{k \sim \hat{\mathcal{K}}} \frac{1}{n_k} \sum_{i=1}^{n_k} l(M_{\boldsymbol{\theta}}, \boldsymbol{z}_{k,i}), \quad (59)$$

where $\hat{\mathcal{K}}$ is an arbitrary distribution on participating nodes that is a part of all nodes, and $R_{\boldsymbol{S}_k}(M_{\boldsymbol{\theta}})$ is the empirical risk for model $M_{\boldsymbol{\theta}}$ on local dataset $\boldsymbol{S}_k$. We further define a partial population risk for model $M_{\boldsymbol{\theta}}$ as

$$R^{\hat{\mathcal{K}}}(M_{\boldsymbol{\theta}}) = \mathbb{E}_{k \sim \hat{\mathcal{K}}} R_k(M_{\boldsymbol{\theta}}) = \mathbb{E}_{k \sim \hat{\mathcal{K}}, \boldsymbol{z} \sim \mathcal{D}_k} l(M_{\boldsymbol{\theta}}, \boldsymbol{z}), \quad (60)$$

where $R_k(M_{\boldsymbol{\theta}})$ is the population risk on node $k$'s data distribution.

Now, we can define the generalization error for dataset $\boldsymbol{S}$ and function $\mathcal{A}(\boldsymbol{S})$ as

$$\Delta_{\mathcal{A}}(\boldsymbol{S}) = R(\mathcal{A}(\boldsymbol{S})) - R_{\boldsymbol{S}}^{\hat{\mathcal{K}}}(\mathcal{A}(\boldsymbol{S})) \quad (61)$$

$$= \underbrace{R(\mathcal{A}(\boldsymbol{S})) - R^{\hat{\mathcal{K}}}(\mathcal{A}(\boldsymbol{S}))}_{\text{Participation gap}} + \underbrace{R^{\hat{\mathcal{K}}}(\mathcal{A}(\boldsymbol{S})) - R_{\boldsymbol{S}}^{\hat{\mathcal{K}}}(\mathcal{A}(\boldsymbol{S}))}_{\text{Out-of-sample gap}}. \quad (62)$$

The expected generalization error is expressed as $\mathbb{E}_{\boldsymbol{S}_k \sim \mathcal{D}k^{n_k}{k=1}^K} \Delta_{\mathcal{A}}(\boldsymbol{S})$. Note that the second term in (62), which is related to the difference between in-sample and out-of-sample loss, can be bounded in the same way as in Theorem 4.1 and Theorem 4.3. The first term is associated with the participation of not all clients. In the following, we demonstrate that under certain conditions, this term would be zero in expectation.

We assume there is a meta-distribution $\mathcal{P}$ supported on all distributions $\hat{K}$.

**Lemma D.1.** *Let $\{x_i\}_{i=1}^{K}$ denote any fixed deterministic sequence. Assume $\mathcal{P}$ is derived by sampling $\hat{K}$ clients with replacement based on distribution $\mathcal{K}$ followed by an equal probability on all sampled clients,* i.e., *$\hat{\mathcal{K}}(k) = \frac{1}{\hat{K}}$. Then*

$$\mathbb{E}_{\mathcal{P}} \mathbb{E}_{k\sim\hat{\mathcal{K}}} x_k = \mathbb{E}_{k\sim\mathcal{K}} x_k, \mathbb{E}_{\mathcal{P}} \mathbb{E}_{k\sim\hat{\mathcal{K}}} \hat{\mathcal{K}}(k)x_k = \frac{1}{\hat{K}} \mathbb{E}_{k\sim\mathcal{K}} x_k, \mathbb{E}_{\mathcal{P}} \mathbb{E}_{k\sim\hat{\mathcal{K}}} \hat{\mathcal{K}}^2(k)x_k = \frac{1}{\hat{K}^2} \mathbb{E}_{k\sim\mathcal{K}} x_k. \tag{63}$$

*Proof.*

$$\mathbb{E}_{\mathcal{P}} \mathbb{E}_{k\sim\hat{\mathcal{K}}} x_k = \mathbb{E}_{\mathcal{P}} \frac{1}{\hat{K}} \sum_{i=1}^{\hat{K}} x_i = \frac{1}{\hat{K}} \sum_{i=1}^{\hat{K}} \mathbb{E}_{\mathcal{P}} x_i = \mathbb{E}_{\mathcal{P}} x_i = \mathbb{E}_{k\sim\mathcal{K}} x_k \tag{64}$$

$$\mathbb{E}_{\mathcal{P}} \mathbb{E}_{k\sim\hat{\mathcal{K}}} \hat{\mathcal{K}}(k)x_k = \mathbb{E}_{\mathcal{P}} \frac{1}{\hat{K}^2} \sum_{i=1}^{\hat{K}} x_i = \frac{1}{\hat{K}^2} \sum_{i=1}^{\hat{K}} \mathbb{E}_{\mathcal{P}} x_i = \frac{1}{\hat{K}} \mathbb{E}_{\mathcal{P}} x_i = \frac{1}{\hat{K}} \mathbb{E}_{k\sim\mathcal{K}} x_k \tag{65}$$

$$\mathbb{E}_{\mathcal{P}} \mathbb{E}_{k\sim\hat{\mathcal{K}}} \hat{\mathcal{K}}^2(k)x_k = \mathbb{E}_{\mathcal{P}} \frac{1}{\hat{K}^3} \sum_{i=1}^{\hat{K}} x_i = \frac{1}{\hat{K}^3} \sum_{i=1}^{\hat{K}} \mathbb{E}_{\mathcal{P}} x_i = \frac{1}{\hat{K}^2} \mathbb{E}_{\mathcal{P}} x_i = \frac{1}{\hat{K}^2} \mathbb{E}_{k\sim\mathcal{K}} x_k \tag{66}$$

$\square$

**Lemma D.2.** *Let $\{x_i\}_{i=1}^{K}$ denote any fixed deterministic sequence. Assume $\mathcal{P}$ is derived by sampling $\hat{K}$ clients without replacement uniformly at random followed by weighted probability on all sampled clients as $\hat{\mathcal{K}}(k) = \frac{\mathcal{K}(k)K}{\hat{K}}$. Then*

$$\mathbb{E}_{\mathcal{P}} \mathbb{E}_{k\sim\hat{\mathcal{K}}} x_k = \mathbb{E}_{k\sim\mathcal{K}} x_k, \mathbb{E}_{\mathcal{P}} \mathbb{E}_{k\sim\hat{\mathcal{K}}} \hat{\mathcal{K}}(k)x_k = \frac{K}{\hat{K}} \mathbb{E}_{k\sim\mathcal{K}} \mathcal{K}(k)x_k, \mathbb{E}_{\mathcal{P}} \mathbb{E}_{k\sim\hat{\mathcal{K}}} \hat{\mathcal{K}}^2(k)x_k = \frac{K^2}{\hat{K}^2} \mathbb{E}_{k\sim\mathcal{K}} \mathcal{K}^2(k)x_k. \tag{67}$$

*Proof.*

$$\mathbb{E}_{\mathcal{P}} \mathbb{E}_{k\sim\hat{\mathcal{K}}} x_k = \mathbb{E}_{\mathcal{P}} \frac{K}{\hat{K}} \sum_{i=1}^{\hat{K}} \mathcal{K}(i)x_i = \frac{K}{\hat{K}} \sum_{i=1}^{\hat{K}} \mathbb{E}_{\mathcal{P}} \mathcal{K}(i)x_i = K \mathbb{E}_{\mathcal{P}} \mathcal{K}(i)x_i = K \frac{1}{K} \sum_{i=1}^{k} \mathcal{K}(i)x_i = \mathbb{E}_{k\sim\mathcal{K}} x_k \tag{68}$$

$$\mathbb{E}_{\mathcal{P}} \mathbb{E}_{k\sim\hat{\mathcal{K}}} \hat{\mathcal{K}}(k)x_k = \mathbb{E}_{\mathcal{P}} \frac{K^2}{\hat{K}^2} \sum_{i=1}^{\hat{K}} \mathcal{K}^2(i)x_i = \frac{K^2}{\hat{K}^2} \sum_{i=1}^{\hat{K}} \mathbb{E}_{\mathcal{P}} \mathcal{K}^2(i)x_i = \frac{K^2}{\hat{K}} \mathbb{E}_{\mathcal{P}} \mathcal{K}^2(i)x_i = \frac{K^2}{\hat{K}} \frac{1}{K} \sum_{i=1}^{k} \mathcal{K}^2(i)x_i \tag{69}$$

$$= \frac{K}{\hat{K}} \mathbb{E}_{k\sim\mathcal{K}} \mathcal{K}(k)x_k$$

$$\mathbb{E}_{\mathcal{P}} \mathbb{E}_{k\sim\hat{\mathcal{K}}} \hat{\mathcal{K}}^2(k)x_k = \mathbb{E}_{\mathcal{P}} \frac{K^3}{\hat{K}^3} \sum_{i=1}^{\hat{K}} \mathcal{K}^3(i)x_i = \frac{K^3}{\hat{K}^3} \sum_{i=1}^{\hat{K}} \mathbb{E}_{\mathcal{P}} \mathcal{K}^3(i)x_i = \frac{K^3}{\hat{K}^2} \mathbb{E}_{\mathcal{P}} \mathcal{K}^3(i)x_i = \frac{K^3}{\hat{K}^2} \frac{1}{K} \sum_{i=1}^{k} \mathcal{K}^3(i)x_i \tag{70}$$

$$= \frac{K^2}{\hat{K}^2} \mathbb{E}_{k\sim\mathcal{K}} \mathcal{K}^2(k)x_k$$

$\square$

So based on lemmas D.1, and D.2, it becomes evident that the expectation of the participation gap in (62) becomes zero for both two methods, *i.e.,*

$$\mathbb{E}_{\mathcal{P}} \left[ R(\mathcal{A}(\boldsymbol{S})) - R^{\hat{\mathcal{K}}}(\mathcal{A}(\boldsymbol{S})) \right] = 0. \tag{71}$$

The expected generalization error, $\mathbb{E}_{\{\boldsymbol{S}_k \sim \mathcal{D}_k^{n_k}\}_{k=1}^{K}} \Delta_{\mathcal{A}}(\boldsymbol{S})$, will be just the expectation of the second term in 62 that can be bounded using Lemma B.2 by

$$\mathbb{E}_{k\sim\hat{\mathcal{K}}} \left[ \frac{L\hat{\mathcal{K}}(k)^2}{\mu} \mathbb{E}_{\{\boldsymbol{S}_k\sim\mathcal{D}_k^{n_k}\}} \Delta_{\mathcal{A}_k}(\boldsymbol{S}_k) + 2\sqrt{\frac{L}{\mu}} \hat{\mathcal{K}}(k) \sqrt{\mathbb{E}_{\{\boldsymbol{S}_k\sim\mathcal{D}_k^{n_k}\}_{k=1}^{K}} \delta_{k,\mathcal{A}}(\boldsymbol{S}) \mathbb{E}_{\{\boldsymbol{S}_k\sim\mathcal{D}_k^{n_k}\}} \Delta_{\mathcal{A}_k}(\boldsymbol{S}_k)} \right]. \tag{72}$$

If we take the expectation of (72) with respect to $\mathcal{P}$, and taking into account Lemma D.1, we get the generalization bound for method 1 as

$$\mathbb{E}_{k\sim\mathcal{K}}\left[\frac{L}{\mu\hat{K}^2}\,\mathbb{E}_{\{\boldsymbol{S}_k\sim\mathcal{D}_k^{n_k}\}}\,\Delta_{\mathcal{A}_k}(\boldsymbol{S}_k) + 2\sqrt{\frac{L}{\mu}}\frac{1}{\hat{K}}\sqrt{\mathbb{E}_{\{\boldsymbol{S}_k\sim\mathcal{D}_k^{n_k}\}_{k=1}^{K}}\,\delta_{k,\mathcal{A}}(\boldsymbol{S})\,\mathbb{E}_{\{\boldsymbol{S}_k\sim\mathcal{D}_k^{n_k}\}}\,\Delta_{\mathcal{A}_k}(\boldsymbol{S}_k)}\right]. \tag{73}$$

For Scheme 2, we can obtain the generalization bound in the same way by taking the expectation of (72) with respect to $\mathcal{P}$ and considering Lemma D.2. We get the generalization bound for Method 2 as:

$$\mathbb{E}_{k\sim\mathcal{K}}\left[\frac{L}{\mu}\frac{\mathcal{K}(k)^2 K^2}{\hat{K}^2}\,\mathbb{E}_{\{\boldsymbol{S}_k\sim\mathcal{D}_k^{n_k}\}}\,\Delta_{\mathcal{A}_k}(\boldsymbol{S}_k) + 2\sqrt{\frac{L}{\mu}}\frac{\mathcal{K}(k)K}{\hat{K}}\sqrt{\mathbb{E}_{\{\boldsymbol{S}_k\sim\mathcal{D}_k^{n_k}\}_{k=1}^{K}}\,\delta_{k,\mathcal{A}}(\boldsymbol{S})\,\mathbb{E}_{\{\boldsymbol{S}_k\sim\mathcal{D}_k^{n_k}\}}\,\Delta_{\mathcal{A}_k}(\boldsymbol{S}_k)}\right]. \tag{74}$$

## E  Detailed Experimental Setup

### E.1  Image Classification

The details are specified in Table 4.

Table 4: Default experimental settings for the image classification training

| | |
|---|---|
| **Dataset** | CIFAR-10, CIFAR-100 (Krizhevsky, 2009), SVHN (Netzer et al., 2011), MNIST (Lecun et al., 1998) |
| **Architecture** | ResNet-20 He et al. (2015) |
| **Training objective** | Cross entropy |
| **Test objective** | Top-1 accuracy |
| **Number of clients** | 5 |
| **Data distribution** | IID (shuffled and split), Non-IID (sorted based on labels then split) |
| **Local Steps $\tau$** | 5 (unless explicitly specified) |
| **Adaptation coefficient $\alpha$** | 10 (unless explicitly specified) |
| **Representation Extractor** | Convolutional layers (unless explicitly specified) |
| **Head** | Final dense layers (unless explicitly specified) |
| **Optimizer** | SGD with momentum |
| **Batch size** | 64 per client |
| **Momentum** | 0.9 (Nesterov) |
| **Learning rate** | Constant $\eta$ after tuning |
| **Number of Iterations** | $10^4$ for IID and $2\times10^4$ for non-IID |
| **Weight decay** | $10^{-4}$ |
| **Repetitions** | 20, with varying seeds |
| **Reported metric** | Mean and standard deviation of the aggregated model's test accuracies over the last 5 rounds |

### E.2 Large Language Model Fine-tuning

The details are specified in Table 5.

Table 5: Default experimental settings for the large language model fine-tuning

| | |
|---|---|
| **Dataset** | Multi-Genre Natural Language Inference (MultiNLI) corpus |
| **Architecture** | OPT-125M He et al. (2015) |
| **Training objective** | Cross entropy |
| **Test objective** | Top-1 accuracy |
| **Number of clients** | 5 |
| **Data distribution** | IID (shuffled and split), |
| | Non-IID (sorted based on genre then split) |
| **Local Steps** $\tau$ | 5 |
| **Adaptation coefficient** $\alpha$ | 10 |
| **Representation Extractor** | First 10 attention layers |
| **Head** | Final 2 attention layers |
| **Optimizer** | AdamW |
| **Batch size** | 16 sentences per client |
| **Adam** $\beta_1$ | 0.9 |
| **Adam** $\beta_2$ | 0.999 |
| **Adam** $\epsilon$ | $10^{-8}$ |
| **Learning rate** | Decaying as $\frac{\eta}{\frac{t}{100}+10}$ where $t$ is the iteration number and $\eta$ is tuned |
| **Number of Iterations** | $10^4$ for IID and $2 \times 10^4$ for non-IID |
| **Weight decay** | $10^{-4}$ |
| **Repetitions** | 20, with varying seeds |
| **Reported metric** | Mean and standard deviation of the aggregated model's test accuracies over the last 5 rounds |

### E.3 Hyper-parameters and tuning details

We independently tuned the learning rates for both image classification and large language model (LLM) fine-tuning experiments for each dataset, algorithm, and data heterogeneity scenario (both iid and non-iid). For image classification, the learning rate was tuned, while for LLM fine-tuning, we specifically tuned the initial learning rate. The tuning process followed these steps:

- Begin with an initial value of 0.01 and perform the experiment several times, each time using a different random seed.

- Perform a grid search to find the optimal learning rate. Start by multiplying or dividing the initial value by powers of two. Test both larger and smaller learning rates, running the experiment multiple times with different random seeds. The best learning rate is the one that, on average, produces the best results. If this learning rate is between two others that give worse results, then you have found the optimal value.

### E.4 Algorithms Used in the Experiments

In this section, we provide a detailed list of our implementation of

- FedAvg (Algorithm 2)

- SCAFFOLD Karimireddy et al. (2019) (Algorithm 3)

- Combined method FedALS+SCAFFOLD (Algorithm 4)

Algorithm 2 demonstrates the implementation of FedAvg. This is a widely used algorithm in federated learning that performs model averaging after local updates from each client.

Moving on to Algorithm 3, we observe that SCAFFOLD introduces an additional mechanism beyond just the model updates. Specifically, SCAFFOLD keeps track of a state that is specific to each client, referred to as the client control variate, denoted by $c_{k,r}$. This control variate is a key feature of SCAFFOLD, which helps to correct the client drift by utilizing these client-specific states. It's important to note that the clients within SCAFFOLD maintain memory and persistently store the values of $c_{k,r}$ and $\sum_{k=1}^{K} c_{k,r}$. This stored information plays a crucial role in the algorithm's functionality. Additionally, it is worth recognizing that when the value of $c_{k,r}$ consistently remains at zero, the SCAFFOLD algorithm essentially reduces to FedAvg. This highlights the close relationship between the two algorithms and emphasizes the special role of the control variates in differentiating SCAFFOLD from FedAvg.

Finally, Algorithm 4 showcases the integration of FedALS with SCAFFOLD, presenting a hybrid approach. It is important to observe that in this combined algorithm, the control variables are not handled in a uniform manner but are instead fragmented according to the different partitions of the model. This fragmentation occurs because there are distinct local step counts for different parts of the model, leading to a more complex update mechanism compared to the standard SCAFFOLD approach.

---

**Algorithm 2** FedAvg

---

**Input**: Initial model $\{\boldsymbol{\theta}_{k,1,0}\}_{k=1}^{K}$, Learning rate $\eta$, and number of local steps $\tau$.
**Output**: $\hat{\boldsymbol{\theta}}_R$

1: **for** Round $r$ in $1, ..., R$ **do**
2:      **for** Node $k$ in $1, ..., K$ **in parallel do**
3:          **for** Local step $t$ in $0, ..., \tau - 1$ **do**
4:             Sample the batch $\mathcal{B}_{k,r,t}$ from $\mathcal{D}_k$.
5:             $\boldsymbol{\theta}_{k,r,t+1} = \boldsymbol{\theta}_{k,r,t} - \frac{\eta}{|\mathcal{B}_{k,r,t}|} \sum_{i \in \mathcal{B}_{k,r,t}} \nabla l(M_{\boldsymbol{\theta}_{k,r,t}}, \boldsymbol{z}_{k,i})$
6:          **end for**
7:          $\boldsymbol{\theta}_{k,r+1,0} = \frac{1}{K} \sum_{k=1}^{K} \boldsymbol{\theta}_{k,r,\tau}$
8:      **end for**
9: **end for**
10: **return** $\hat{\boldsymbol{\theta}}_R = \frac{1}{K} \sum_{k=1}^{K} \boldsymbol{\theta}_{k,R,\tau}$

---

---

**Algorithm 3** SCAFFOLD

---

**Input**: Initial model $\{\boldsymbol{\theta}_{k,1,0}\}_{k=1}^{K}$, Initial control variable $\{\boldsymbol{c}_{k,1}\}_{k=1}^{K}$, learning rate $\eta$, and number of local steps $\tau$.

**Output**: $\hat{\boldsymbol{\theta}}_R$

1: **for** Round $r$ in $1, ..., R$ **do**
2:    **for** Node $k$ in $1, ..., K$ **in parallel do**
3:       **for** Local step $t$ in $0, ..., \tau - 1$ **do**
4:          Sample the batch $\mathcal{B}_{k,r,t}$ from $\mathcal{D}_k$.
5:          $\boldsymbol{\theta}_{k,r,t+1} = \boldsymbol{\theta}_{k,r,t} - \eta\big(\frac{1}{|\mathcal{B}_{k,r,t}|} \sum_{i \in \mathcal{B}_{k,r,t}} \nabla l(M_{\boldsymbol{\theta}_{k,r,t}}, \boldsymbol{z}_{k,i}) - \boldsymbol{c}_{k,r} + \frac{1}{K} \sum_{k=1}^{K} \boldsymbol{c}_{k,r}\big)$
6:       **end for**
7:       $\boldsymbol{c}_{k,r+1} = \boldsymbol{c}_{k,r} - \frac{1}{K} \sum_{k=1}^{K} \boldsymbol{c}_{k,r} + \frac{1}{\eta\tau}(\boldsymbol{\theta}_{k,r,0} - \boldsymbol{\theta}_{k,r,\tau})$
8:       $\boldsymbol{\theta}_{k,r+1,0} = \frac{1}{K} \sum_{k=1}^{K} \boldsymbol{\theta}_{k,r,\tau}$
9:    **end for**
10: **end for**
11: **return** $\hat{\boldsymbol{\theta}}_R = \frac{1}{K} \sum_{k=1}^{K} \boldsymbol{\theta}_{k,R,\tau}$

---

**Algorithm 4** FedALS + SCAFFOLD

---

**Input**: Initial model $\{\boldsymbol{\theta}_{k,1,0} = [\boldsymbol{\phi}_{k,1,0}, \boldsymbol{h}_{k,1,0}]\}_{k=1}^{K}$, Initial control variable $\{\boldsymbol{c}_{k,1} = [\boldsymbol{c}_{k,1}^{\boldsymbol{\phi}}, \boldsymbol{c}_{k,1}^{\boldsymbol{h}}]\}_{k=1}^{K}$, learning rate $\eta$, number of local steps for the head model $\tau$, adaptation coefficient $\alpha$.

**Output**: $\hat{\boldsymbol{\theta}}_R$

1: **for** Round $r$ in $1, ..., R$ **do**
2:    **for** Node $k$ in $1, ..., K$ **in parallel do**
3:       **for** Local step $t$ in $0, ..., \tau - 1$ **do**
4:          Sample the batch $\mathcal{B}_{k,r,t}$ from $\mathcal{D}_k$.
5:          $\boldsymbol{\theta}_{k,r,t+1} = \boldsymbol{\theta}_{k,r,t} - \eta\big(\frac{1}{|\mathcal{B}_{k,r,t}|} \sum_{i \in \mathcal{B}_{k,r,t}} \nabla l(M_{\boldsymbol{\theta}_{k,r,t}}, \boldsymbol{z}_{k,i}) - \boldsymbol{c}_{k,r} + \frac{1}{K} \sum_{k=1}^{K} \boldsymbol{c}_{k,r}\big)$
6:          **if** $\mod(r\tau + t, \tau) = 0$ **then**
7:             $\boldsymbol{c}_{k,r}^{\boldsymbol{h}} \leftarrow \boldsymbol{c}_{k,r}^{\boldsymbol{h}} - \frac{1}{K} \sum_{k=1}^{K} \boldsymbol{c}_{k,r}^{\boldsymbol{h}} + \frac{1}{\eta\tau}(\boldsymbol{h}_{k,r,0} - \boldsymbol{h}_{k,r,t})$
8:             $\boldsymbol{h}_{k,r,t} \leftarrow \frac{1}{K} \sum_{k=1}^{K} \boldsymbol{h}_{k,r,t}$
9:          **end if**
10:         **if** $\mod(r\tau + t, \alpha\tau) = 0$ **then**
11:            $\boldsymbol{c}_{k,r}^{\boldsymbol{\phi}} \leftarrow \boldsymbol{c}_{k,r}^{\boldsymbol{\phi}} - \frac{1}{K} \sum_{k=1}^{K} \boldsymbol{c}_{k,r}^{\boldsymbol{\phi}} + \frac{1}{\eta\alpha\tau}(\boldsymbol{\phi}_{k, \lfloor\frac{r\tau+t-\alpha\tau}{\tau}\rfloor, \mod(r\tau+t-\alpha\tau, \tau)}^{l} - \boldsymbol{\phi}_{k,r,t}^{l})$
12:            $\boldsymbol{\phi}_{k,r,t} \leftarrow \frac{1}{K} \sum_{k=1}^{K} \boldsymbol{\phi}_{k,r,t}$
13:          **end if**
14:       **end for**
15:       $\boldsymbol{c}_{k,r+1} = \boldsymbol{c}_{k,r}$
16:       $\boldsymbol{\theta}_{k,r+1,0} = \boldsymbol{\theta}_{k,r,\tau}$
17:    **end for**
18: **end for**
19: **return** $\hat{\boldsymbol{\theta}}_R = \frac{1}{K} \sum_{k=1}^{K} \boldsymbol{\theta}_{k,R,\tau}$

---

