# OpenReview forum: "Differentiated Aggregation to Improve Generalization in Federated Learning"
_TMLR — Accepted by TMLR_

### Review · Reviewer_7HKh · 2025-05-08

**Summary Of Contributions:**

This paper conducts a detailed analysis of the communication cost of federated learning algorithms, specifically FedAvg, in terms of generalization. Based on their analysis, the authors observe that the representations across clients learn more shared features in early layers than final layers, and accordingly propose a new algorithm (FedALS) which adaptively reduces the aggregation frequency of local weights for those layers. Through extensive experimentation, the authors demonstrate that FedALS leads to better generalization with reduced communication overhead.

**Audience:**

Yes

**Claims And Evidence:**

Yes

**Requested Changes:**

See above.

**Strengths And Weaknesses:**

1) The paper carefully separates the contributions coming from non-IIDness to demonstrate that the previous error bounds 1/K improve to 1/K^2 when the dataset across all nodes is uniformly distributed.

- In Eq.9, the abstract model complexity's appearance obscures the bound's practical applications. It would be helpful to demonstrate the tightness of the bound, at least for synthetic data scenarios.

2) The FedALS algorithm is empirically well-motivated by the observation that early layers should learn global features rather than head layers, which are more task-specific. The algorithm is easy to implement and shows superior performance to its predecessors.

- The bounds depend on strong assumptions on convexity and smoothness, which are not guaranteed in deep neural nets. While FedALS in demonstrated examples performs better than previous methods, it is unclear whether this will hold in other applications. It would be helpful to present a numerical test of how predictive the theory is for at least synthetic data.

- The representation learning perspective is mainly justified based on empirical observations. I believe this narrative can be strengthened by theoretical studies that demonstrate that the representations in earlier layers align with the input data and final layers with the task structure. Here are a few works that I know of:
    - https://arxiv.org/pdf/2012.04030
     - https://arxiv.org/abs/2106.00651
    - https://arxiv.org/abs/2205.09653

- Beyond expensive hyperparameter searches, is there a principled way to choose $\alpha$ based on theory? What are some factors that $\alpha$ strongly depends on?

Overall, I believe this is a solid paper, both deriving stronger bounds for federated learning and introducing a novel algorithm. It provides valuable insight into federated learning and is clearly written. I think a deeper and clearer discussion of its theoretical assumptions, experimental guidance, and its limitations would strengthen the paper further.

---

> ### Author Response · Authors · 2025-07-11
> **Response to Reviewer 7HKh**
>
> We would like to thank you for taking the time to read our submission and for providing valuable comments that contributed significantly to the improvement of the paper.
>
> Our point-by-point response, including figures, can be found at the following link:
>
> https://drive.google.com/file/d/1i9KJgBV5lOmuABa9UDMIa6PctmFvUJTP/view?usp=share_link
>
> **Numerical test of the theory:**
> To validate Theorem 4.1, we conduct numerical experiments on a synthetic dataset where data samples are drawn from a Gaussian distribution. The theorem provides a generalization bound based on local generalization and data noniid-ness, and we test its predictions using both convex and non-convex models.
> First, we use a logistic regression model, which has a convex loss function that satisfies the assumptions in Theorem 4.1. As shown in Figure 1a in the link, the results demonstrate a quadratic improvement in both the generalization error and its theoretical upper bound.
> Next, we test a non-convex two-layer perceptron model. In this case, Figure 1b in the link shows that the quadratic ($\frac{1}{K^2}$) behavior is no longer observed, which is expected as the convexity assumption is violated. However, the bound provided by Theorem 4.1 remains remarkably tight, demonstrating its practical utility even beyond its strict theoretical requirements.
>
> **Theoretical studies:**
> We want to thank the reviewer for their insightful comment. We will include the following parapgraph in the introduction section of the paper to strengthen our argument:
>
> _This view is supported by recent theoretical work showing that as you go deeper into a network, the way it characterizes information changes in a structured way. For example, [2] and [3] have found that a layer's characterization is a blend of the input data's structure and the learning task's structure. Critically, they showed that the influence of the task becomes stronger in deeper layers, meaning early layers focus more on general input features while later layers adapt to focus on what’s needed to solve the task. Also [1], by doing a study on learning dynamics, confirmed this, showing that deeper layers change their characterization more significantly during training to ultimately align with the final goal. Taken together, these papers provide a strong theoretical basis for the idea that networks learn by gradually shifting their focus from representing the input data in early layers to characterizing the task solution in later layers._
>
> **hyperparameter searches for $\alpha$:**
> Theoretically, the optimal value of $\alpha$ depends on the discrepancy in how noniid data affects the representation extractor versus the head. As we demonstrated in this paper, increasing the number of local steps for the representation extractor improves generalization. This is not necessarily true for the head, which is more sensitive to the noniid nature of the local data.
> Quantifying this discrepancy is challenging, as it depends on factors like the model's class complexity and the precise nature of the data distribution across clients. For this reason, we treat $\alpha$ as a hyperparameter. However, this hyperparameter is not unconstrained. It represents an optimal trade-off within a well-defined range, starting from $\alpha=1$ (equivalent to standard FedAvg) to $\alpha=R$ (representing one-shot averaging of the representations at the end of training). Given this bounded range, the optimal $\alpha$ can be found efficiently using methods like binary search to a desired precision. For our experiments, we found that a search with integer precision was sufficient.
>
> We are happy to clarify any further points.
>
> Thank You!
>
> -------
> [1] Bordelon, Blake, and Cengiz Pehlevan. "Self-consistent dynamical field theory of kernel evolution in wide neural networks." Advances in Neural Information Processing Systems 35 (2022): 32240-32256..
>
> [2]Li, Qianyi, and Haim Sompolinsky. "Statistical mechanics of deep linear neural networks: The backpropagating kernel renormalization." Physical Review X 11.3 (2021): 031059.
>
> [3]Zavatone-Veth, Jacob, et al. "Asymptotics of representation learning in finite Bayesian neural networks." Advances in neural information processing systems 34 (2021): 24765-24777.

---

### Review · Reviewer_cj4e · 2025-05-12

**Summary Of Contributions:**

This paper presents a novel and tighter generalization bound for FL in the non-iid. setting and, together with empirical findings from the field of representation learning, propose a new FL algorithm that reduces communication efficiency while improving performance (test loss and accuracy). In particular, they find that their derived generalization bound can be decreased by decreasing the aggregation frequency of the feature extractor, as compared to the classification head. This, in turn, increases the number of samples and local SGD steps terms in the bound, while keeping the increase of the non-iidness term comparatively low, which ultimately decreases the derived bound.

**Audience:**

Yes

**Broader Impact Concerns:**

If it turns out that the method doesn’t work for a larger number of clients, a limitations section should be added that explains that the method is only applicable to the cross-silo FL setting [6] with full client participation.



[6] Kairouz, Peter, et al. "Advances and open problems in federated learning." Foundations and trends® in machine learning 14.1–2 (2021): 1-210.

**Claims And Evidence:**

Yes

**Requested Changes:**

- Add more baselines, e.g.: FedProx [2], FedOpt [1].
- Test the method on a larger number of clients, e.g., 100, as is commonly done [3].
- As an ablation study experiment, vary the fraction of clients used for training in each round, e.g. 10 out of 100, randomly sampled.
- As an ablation study experiment, vary
- While in Figure 2 you plot (from left to right):
Training global loss (non-iid), Test accuracy (non-iid), Training global loss (iid.), Training accuracy (iid.), in Figure 3 you plot: Training global loss (non-iid), Test loss (non-iid), Training global loss (iid.), Training accuracy (iid.). Why? Please plot, from left to right:
Training global loss (non-iid), Test accuracy (non-iid), Training global loss (iid.), Training accuracy (iid.) for both the CNN and the LLM. if you want to add test loss (non-iid), add another column.
- Tables 2 & 3 include ablation studies for L and alpha only for the CNN. Please run the same ablation study experiments with the LLM.
- For the non-iid experiments, add the commonly used Dirichlet label split method [4] for your experiments and real-life FL splits, such as FEMNIST [5].

[1] Reddi, Sashank, et al. "Adaptive federated optimization." arXiv preprint arXiv:2003.00295 (2020).

[2] Li, Tian, et al. "Federated optimization in heterogeneous networks." Proceedings of Machine learning and systems 2 (2020): 429-450.

[3] McMahan, Brendan, et al. "Communication-efficient learning of deep networks from decentralized data." Artificial intelligence and statistics. PMLR, 2017.

[4] Yurochkin, Mikhail, et al. "Bayesian nonparametric federated learning of neural networks." International conference on machine learning. PMLR, 2019.

[5] Caldas, Sebastian, et al. "Leaf: A benchmark for federated settings." arXiv preprint arXiv:1812.01097 (2018).

[6] Kairouz, Peter, et al. "Advances and open problems in federated learning." Foundations and trends® in machine learning 14.1–2 (2021): 1-210.

**Strengths And Weaknesses:**

Strengths:
Theoretically sound derivation of a new FL method that decreases generalization error while also decreasing the communication overhead is a crucial objective in FL.

Weaknesses:
More experiments are needed to verify the proposed method empirically. In particular, it is not clear whether the method will work on a larger set of clients and how it behaves when the fraction of clients used in each round for training varies. Note, this is commonly analyzed in FL. Moreover, there is no information on whether the method can compete with other popular global FL methods, e.g. FedOpt [1] etc.

[1] Reddi, Sashank, et al. "Adaptive federated optimization." arXiv preprint arXiv:2003.00295 (2020).

---

> ### Author Response · Authors · 2025-07-11
> **Response to Reviewer cj4e**
>
> We would like to thank you for taking the time to read our submission and for providing valuable comments that contributed significantly to the improvement of the paper.
>
> Our point-by-point response, including figures, can be found at the following link:
>
> https://drive.google.com/file/d/1JJ_sUhU0oEv6Irt5twYU2EKcy9_oRAY1/view?usp=share_link
>
>
> **Response to weaknesses and requested changes:**
> Following the reviewer's suggestions, we conducted additional experiments to further validate our method. These results are available at the link above.
>
> We are happy to clarify any further points.
>
> Thank You!

---

### Review · Reviewer_MdNz · 2025-07-18

**Summary Of Contributions:**

Comment 1: The paper's theoretical motivation is contradictory. The authors claim in Section 5 that increasing local steps (τ) improves the generalization bound in Theorem 4.3, yet they also state it increases non-iidness (δk,A), a term that worsens the same bound. The paper must provide a rigorous analysis that formally reconciles this tension to justify its core theoretical claim.

Comment 2: The method for splitting the model into a "representation extractor" and a "head" is presented as a critical design choice in Algorithm 1 but is treated as an arbitrary hyperparameter in the experiments (Section 6 and Table 3). For FedALS to be a generalizable method, the authors must provide a principled, systematic procedure for determining this architectural split, rather than relying on model-specific ad-hoc tuning.

Comment 3: The experimental comparison to SCAFFOLD in Section 6.3 is misleadingly framed as competitive rather than complementary. The fact that "FedALS + SCAFFOLD" achieves the best performance in Table 1 strongly indicates the two methods address orthogonal problems. The paper should reframe this discussion to highlight this synergy, positioning FedALS as a communication-efficiency method that can be combined with variance-reduction techniques.

Comment 4: The empirical analysis of the key hyperparameter α in Table 2 is disconnected from the paper's own theoretical framework. The authors observe that performance degrades for α > 10 due to non-iidness but miss the crucial opportunity to connect this finding back to the generalization bound in Theorem 4.3. A stronger analysis would empirically estimate the bound's components to show how the theory explains this observed performance peak.

Comment 5: The core motivation for FedALS—that initial layers have lower consensus distance—is only empirically validated for the ResNet-20 model in Figure 1. The paper then applies the same logic to fine-tuning an OPT-125M model in Section 6 without providing any evidence that its initial layers exhibit the same property. This unsubstantiated assumption weakens the justification for applying FedALS to LLMs.

**Audience:**

Yes

**Claims And Evidence:**

Yes

**Requested Changes:**

N/A

**Strengths And Weaknesses:**

Strengths: The paper's core contribution is FedALS, a novel and intuitive algorithm that improves both accuracy and communication efficiency in non-iid federated learning. The key insight—to aggregate the model's initial "representation extractor" layers less frequently than its final "head" layers—is well-motivated by both a theoretical generalization bound analysis (Section 4) and compelling empirical evidence showing lower consensus distance in early layers (Figure 1). The experimental results in Table 1 and Table 2 strongly validate this approach, demonstrating that FedALS achieves higher final accuracy while simultaneously reducing communication overhead.

Limitations: The primary weakness is the paper's contradictory theoretical justification for its core idea. In Section 5, the authors claim that increasing the number of local steps (τ) tightens their generalization bound in Theorem 4.3, which motivates the FedALS algorithm. However, the same theorem includes a non-iidness term (δk,A) that, as the authors acknowledge, worsens with increased τ. The paper fails to formally analyze or resolve this critical trade-off within its own theoretical framework, which undermines the claim that the algorithm's design is directly and rigorously derived from the provided bounds.

---

> ### Author Response · Authors · 2025-07-25
> **Response to Reviewer MdNz**
>
> We thank the reviewer for their insightful feedback, which has helped us strengthen the paper. The following
> is our point-to-point reply.
> Our response, including figures, can be found here: https://drive.google.com/file/d/1MJ8np3-gxGuYiffv4k89_7X80mbZPlw1/view?usp=share_link
>
> **Response to Comment 1:**
> We thank the reviewer for this insightful question. The  ``contradiction'' the reviewer mentioned is, in fact, the fundamental trade-off that our paper aims to leverage.
>
> Increasing the number of local steps ($\tau$) creates a trade-off within the bound in Theorem 4.3: It improves the bound by reducing Term A and the second component of Term B ($\frac{\sigma^2}{\mu \tau}$), but worsens it by increasing the first component of Term B, which grows with noniid-ness.
>
> An exact analysis relies on the behavior of the complex term of $\delta_{k,\mathcal{A}} (\cdot)$, which is difficult to characterize without making strong assumptions about the data distribution and loss function.  As a result, a more precise analysis remains currently out of reach.
>
> Our key insight is that this trade-off can be managed by recognizing that different parts of the model have different sensitivities to noniid-ness. Based on prior work [1-3] and our experiments (Figure 1 in the main manuscript), the feature extractor is less affected by noniid data than the head. Consequently, our differential strategy is to apply a larger $\tau$ to the extractor to gain its benefits and a smaller $\tau$ to the head to avoid the large noniid penalty. This approach is our proposed reconciliation of the tension.
>
> **Response to Comment 2:**
> We thank the reviewer for this important question. We agree that a principled procedure for the model split is essential for generalizability. Our approach is guided by established theoretical work on representation learning and is not an arbitrary hyperparameter.
>
> The basis for our split is the functional characterization of layers in a neural network.
> Theoretical studies demonstrate a fundamental principle: neural networks' early layers learn general features, and subsequent layers then adapt to the specific task. This pattern is consistent across different inputs and tasks [1-3]. This provides the theoretical foundation for separating a robust ``representation extractor`` from a task-specific ``head.``
>
> While this principle provides a clear basis for the split, the precise architectural boundary where this functional role transitions is not typically known \textit{a priori} and can vary between model architectures. Consequently, we employ an empirical approach to determine this point.
>
> To support this, we performed experiments on the ResNet-20 architecture across three different datasets. As shown in Table 3, the optimal split point was consistent across all three datasets. This consistency demonstrates that our method identifies a fundamental and stable characteristic of the model architecture itself.
>
> **Response to Comment 3:**
> We agree with the reviewer's insightful point. Methods like SCAFFOLD and FedProx are indeed complementary to FedALS. They can be applied concurrently to gain the dual benefits of improved communication efficiency from FedALS while further combating the effects of non-iid data. We will updated Section 6.3 to highlight this synergy.
>
>
> --------------
> [1] Bordelon, Blake, and Cengiz Pehlevan. "Self-consistent dynamical field theory of kernel evolution in wide neural networks." Advances in Neural Information Processing Systems 35 (2022): 32240-32256..
>
> [2]Li, Qianyi, and Haim Sompolinsky. "Statistical mechanics of deep linear neural networks: The backpropagating kernel renormalization." Physical Review X 11.3 (2021): 031059.
>
> [3]Zavatone-Veth, Jacob, et al. "Asymptotics of representation learning in finite Bayesian neural networks." Advances in neural information processing systems 34 (2021): 24765-24777.

---

> ### Author Response · Authors · 2025-07-25
> **Response to Reviewer MdNz (Continued)**
>
> **Response to Comment 4:**
> The performance degradation observed for $\alpha > 10$ in Table 2 is indeed explained by our theoretical framework, specifically by the behavior of the noniid-ness term, $\delta_{K,\mathcal{A}}$.
>
> While FedALS relies on the fact that the representation extractor is more robust to noniid data than the head, its robustness is not infinite. Using a very large value for $\alpha$ (i.e., many local steps) causes the noniid-ness of the representation extractor itself to grow and eventually degrade the overall performance.
>
> Note that $\delta_{k,{A}}(S_k) = R_{s_k}({A}(S_k)) -  R_{s_k}({A_k}(S_k))$. By increasing $\alpha$, we increase the number of local steps for the extractor. This pushes the local extractor, closer to the optimum of its own local data, causing it to diverge from the global model. As this divergence grows, the noniid-ness term $\delta_{k,\mathcal{A}}$ becomes large, and its negative impact on the generalization bound eventually outweighs the benefits of more local training. This theoretical trade-off is precisely what leads to the performance peak observed empirically in our experiments.
>
>
> **Response to Comment 5:**
> We thank the reviewer. To address this point, we have performed the same consensus distance analysis on the OPT-125M model that was previously done for ResNet-20. The results of this new experiment are presented in Figure1 in the provided link.
>
> Consistent with our findings for ResNet-20, we observe that the initial transformer layers of OPT-125M show a higher degree of consensus (i.e., lower distance) across clients compared to the deeper, task-specific layers.
>
> **Response to Limitations:**
> Please refer to **Response to Comment 1**.
>
> We are happy to clarify any further points.
>
> Thank You!

---

### Decision · Action_Editor_vwj2 · 2025-10-12

**Recommendation:** Accept as is

**Audience:**

Yes

**Audience Explanation:**

The paper targets Federated Learning Community

**Claims And Evidence:**

Yes

**Claims Explanation:**

There was a consensus among the reviewers that the claims made by the authors are supported. The authors addressed most of the concerns raised by the reviewers and the reviewers were satisfied with the additional information and experiments provided by the authors.